# Rh-3DGS: Robust Open-Vocabulary Scene Understanding via Riemannian Huber Distillation and Manifold-Aware Sampling

**Xinpeng Zhao** [1 2]  **Jie Jiang** [1 2]  **Fengyuan Zhang** [3]  **Lixin Zhan** [1 2]  **Dong Wang** [1 2]  **Qinyuan Bu** [4]  **Jiahang Tu** [4]  **Guangzhen Yao** [4]

## Abstract

Open-vocabulary 3D scene understanding answers free-form text queries over reconstructed scenes. However, lifting dense 2D foundation-model embeddings into 3D Gaussian Splatting (3DGS) is still challenging. Existing 3DGS-based methods often average normalized embeddings in Euclidean space. This ignores their hyperspherical geometry and can cause feature collapse. They also distill supervision from all views equally, which amplifies occlusion noise and mixed-depth artifacts. We propose **Rh-3DGS**, a robust semantic 3DGS framework that uses reliability-aware distillation and manifold-consistent aggregation. **Visibility-Calibrated Distillation (VCD)** computes per-pixel reliability weights from rasterization statistics and down-weights ambiguous pixels. **Visibility-Weighted Fréchet Mean (VFM)** aggregates embeddings on the unit hypersphere with a Riemannian Huber objective for robust distillation. **Lightweight Consistency Contrast (LIC)** regularizes the 3D semantic field with neighborhood-based multi-positive contrast to improve local consistency and sharper boundaries. Experiments on three benchmarks show that Rh-3DGS is best on open-vocabulary segmentation, boundary quality, and view-consistent rendering.

## 1. Introduction

Recent progress in explicit 3D representations has enabled high-quality and real-time rendering. A prominent exam-

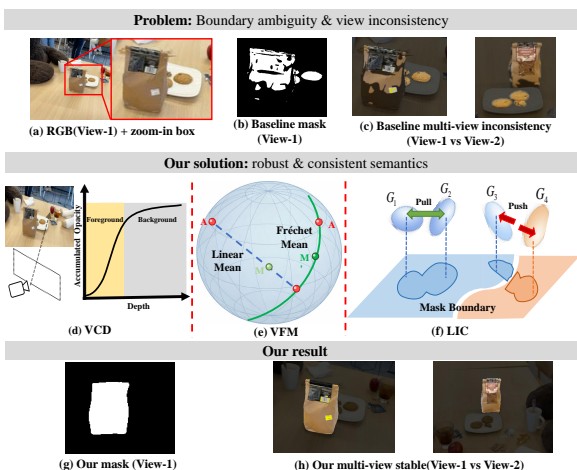

*Figure 1.* **Motivation on LERF (teatime, "bag of cookies").** Baseline 3DGS produces boundary bleeding and multi-view inconsistent masks under occlusion and mixed-depth rays (b–c). Rh-3DGS applies VCD for pixel reliability, VFM for hyperspherical aggregation, and LIC for local 3D consistency, yielding sharper and view-stable masks (g–h).

ple is 3D Gaussian Splatting (3DGS) ([Kerbl et al., 2023](#)), which reconstructs photo-realistic scenes and renders efficiently ([Yu et al., 2021](#)). However, vanilla 3DGS is **semantic-agnostic**. For robotics, scene editing, and embodied assistants, rendering is not enough. The system must also understand scene semantics ([Fei et al., 2024](#)).

A key trend in vision is the shift from closed-set labels to open-vocabulary understanding, especially in 3D ([Zhu & Chen, 2024](#); [Zhan et al., 2026](#)). Classic 3D semantic segmentation assumes a fixed label set. This limits real-world use because new concepts keep appearing. Recent open-vocabulary 3D methods lift representations from large pre-trained 2D vision–language models (VLMs), such as CLIP ([Radford et al., 2021](#)), into 3D. This enables text-driven querying and segmentation in reconstructed scenes.

Despite this progress, lifting 2D open-vocabulary features to 3D Gaussians is still difficult. A main reason is that existing pipelines ignore the **geometry of semantic embeddings**. Most methods ([Qin et al., 2024](#); [Wu et al., 2024](#); [Szilagyi et al., 2025](#)) treat teacher embeddings as Euclidean

---

[1]College of Systems Engineering, National University of Defense Technology, China [2]Laboratory for Big Data and Decision, National University of Defense Technology, China [3]College of Electronic Science and Technology, National University of Defense Technology,China [4]College of Science, National University of Defense Technology,China. Correspondence to: Jie Jiang <jiejiang@nudt.edu.cn>.

*Proceedings of the 43rd International Conference on Machine Learning*, Seoul, South Korea. PMLR 306, 2026. Copyright 2026 by the author(s).

*Table 1.* **Empirical evidence of the Euclidean–hyperspherical mismatch on the baseline model.**

| Region | Pre-norm $\|s\|_2$ | Ang. dev. |
|---|---|---|
| All | 0.957 | 16.16 |
| Bnd. | 0.924 | 18.12 |
| Non-bnd. | 0.965 | 15.66 |
| High-var | 0.945 | 17.66 |
| Low-var | 0.960 | 15.77 |

vectors and fuse them by linear interpolation or averaging. This is not geometry-consistent. Pre-trained embeddings are usually $\ell_2$-normalized and lie on (or near) a unit hypersphere (Liu et al., 2017; Zhan et al., 2025). Euclidean averaging shrinks feature norms and pulls embeddings toward the origin. This causes **feature collapse** and reduces discriminability (Figure 1(e)).

Beyond the qualitative illustration in Fig. 1, we directly measure this mismatch on the baseline model. As shown in Tab. 1, Euclidean fusion yields both norm shrinkage and directional distortion in practice. More importantly, both effects are amplified on boundary and high-depth-variance pixels, exactly the regions where semantic supervision is least reliable. This provides direct empirical evidence for the geometric motivation of our manifold-aware distillation design.

A second issue is the quality of multi-view supervision. Many pipelines distill from all pixels equally. But pixels near occlusion boundaries are often unreliable. Alpha compositing can also mix multiple depths along one ray. Such pixels provide noisy and view-dependent supervision. This leads to blurry boundaries and unstable masks across views. Figure 1(d) illustrates this reliability gap.

To address these issues, we propose **Rh-3DGS**, a robust open-vocabulary scene understanding framework built on 3DGS. Rh-3DGS combines *geometry-aware supervision* and *manifold-consistent aggregation*. It introduces three components. (1) **Visibility-Calibrated Distillation (VCD)** computes pixel-wise reliability weights from rasterization statistics and down-weights occluded and mixed-depth regions. (2) **Visibility-Weighted Fréchet Mean (VFM)** aggregates features directly on the hypersphere and prevents collapse through robust Riemannian distillation. (3) **Lightweight Consistency Contrast (LIC)** enforces local semantic consistency in 3D neighborhoods. It is most helpful near boundaries, where local contrast reduces semantic bleeding (Figure 1(f)).

Our contributions are summarized as follows:

- We identify the *manifold inconsistency* in prior open-vocabulary 3DGS pipelines, and provide direct empirical evidence that Euclidean fusion of normalized embeddings causes measurable norm shrinkage and angular distortion.

- We develop a robust distillation framework that combines **VCD**, **VFM**, and **LIC** to improve semantic reliability, discriminability, and multi-view consistency in 3D Gaussian representations.

- We evaluate Rh-3DGS on diverse open-vocabulary 3D benchmarks, and show consistent improvements over strong baselines in segmentation quality and view-consistent rendering.

## 2. Related Work

### 2.1. 3D Gaussian Splatting for Efficient Neural Rendering

Neural Radiance Fields (NeRF) (Mildenhall et al., 2021) achieve high-quality novel view synthesis. They optimize an implicit radiance field. However, NeRF relies on volumetric ray marching, which is slow for interactive use. To improve efficiency, recent methods use explicit scene representations. 3D Gaussian Splatting (3DGS) (Kerbl et al., 2023) represents a scene as anisotropic Gaussians. Each Gaussian has a center, covariance, opacity, and view-dependent appearance (e.g., SH coefficients). A fast differentiable rasterizer renders these Gaussians. This design enables real-time rendering and fast optimization with strong visual quality. However, vanilla 3DGS focuses on geometry and photometry. It does not model semantic fields for downstream understanding.

### 2.2. Open-Vocabulary 3D Scene Understanding

Open-vocabulary 3D understanding aims to recognize and segment arbitrary categories. It transfers semantics from large vision–language models (VLMs), such as CLIP (Radford et al., 2021) and LSeg (Li et al., 2022). Early works lift 2D embeddings into NeRFs or 3D primitives by aggregating multi-view observations. Examples include LERF (Kerr et al., 2023) and OpenScene (Peng et al., 2023). These NeRF-based pipelines are still costly at inference. Recent work brings open-vocabulary lifting to 3DGS. A common approach is to attach semantic features to Gaussians and train them with distillation (e.g., LangSplat (Qin et al., 2024), Feature-3DGS (Zhou et al., 2024)). Most methods fuse per-view embeddings with Euclidean averaging or alpha blending. This assumes a flat feature space. In practice, many VLM embeddings are $\ell_2$-normalized and lie on $\mathbb{S}^{D-1}$. Euclidean fusion can shrink norms and distort angular relations. This motivates manifold-aware aggregation, such as Riemannian averaging and Fréchet means (Le, 2001; Lim & Pálfia, 2012).

## 2.3. Semantic Distillation in 3D Gaussian Splatting

Many 3DGS semantic methods learn per-Gaussian features from 2D teachers. Feature-3DGS (Zhou et al., 2024) distills dense segmentation embeddings (e.g., LSeg) into compact features. LangSplat (Qin et al., 2024) compresses high-dimensional CLIP features with an autoencoder for efficient rasterization. These methods show strong results, but two issues remain.

(*i*) **Noisy and view-dependent supervision:** 2D semantic maps can change across views. Occlusions, scale changes, and boundary mixing cause inconsistencies (Wang et al., 2025; Tian et al., 2025; Lee et al., 2025; Umam et al., 2024). If we distill all pixels equally, the 3D field becomes blurry near boundaries. This motivates geometry-aware reliability weighting during distillation (Wang et al., 2025; Qiu et al., 2024).

(*ii*) **Mismatch to feature geometry:** Alpha blending performs Euclidean interpolation. This conflicts with normalized embeddings on $\mathbb{S}^{D-1}$. Work on Riemannian statistics and robust estimation suggests using intrinsic distances and manifold means for spherical data (Sato, 2021).

## 2.4. Contrastive Learning and Multi-view Consistency

Contrastive learning and consistency regularization improve dense representations. They often sharpen decision boundaries and reduce noise (Wu et al., 2021; 2018). In 3D and multi-view settings, neighborhood sampling and cross-view constraints are also effective. They help suppress view-dependent artifacts (Tulsiani et al., 2018; Tang et al., 2025). These ideas motivate lightweight contrastive regularization on 3D primitives.

## 3. Preliminaries

### 3.1. 3D Gaussian Splatting

3DGS (Kerbl et al., 2023) represents a scene as anisotropic Gaussians $\mathcal{G} = \{g_i\}_{i=1}^N$ with parameters $(\boldsymbol{\mu}_i, \boldsymbol{\Sigma}_i, \alpha_i, \mathbf{c}_i)$. For a pixel $\mathbf{u}$, let $\mathcal{N}(\mathbf{u})$ be the set of contributing Gaussians in front-to-back order. 3DGS renders via transmittance-weighted $\alpha$-compositing:

$$\mathbf{C}(\mathbf{u}) = \sum_k T_k(\mathbf{u}) \, \alpha'_k(\mathbf{u}) \, \mathbf{r}_k(\mathbf{u}),$$
$$T_k(\mathbf{u}) = \prod_{m<k} \big(1 - \alpha'_m(\mathbf{u})\big). \tag{1}$$

### 3.2. Feature Rendering and Distillation

**Feature rendering.** A common practice in semantic 3DGS is to associate each Gaussian with a learnable semantic descriptor in a teacher feature space and render a feature map using the same transmittance-weighted $\alpha$-compositing as RGB:

$$\mathbf{F}_{\text{render}}(\mathbf{u}) = \sum_k T_k(\mathbf{u}) \, \alpha'_k(\mathbf{u}) \, \mathbf{h}_k, \tag{2}$$

where $\mathbf{h}_k \in \mathbb{R}^D$ denotes the decoded semantic feature of the $k$-th Gaussian.

**Our parameterization.** Instead of directly optimizing $\mathbf{h}_i$ as free $D$-dim vectors, we store a low-dimensional semantic latent $\mathbf{f}_i \in \mathbb{R}^d$ for memory efficiency and decode it by a lightweight decoder: $\mathbf{h}_i = \text{Dec}(\mathbf{f}_i) \in \mathbb{R}^D$. For hyperspherical distillation, we normalize teacher and rendered embeddings at the pixel level (e.g., $\tilde{\mathbf{F}}_{\text{render}}(\mathbf{u}) = \mathbf{F}_{\text{render}}(\mathbf{u})/\|\mathbf{F}_{\text{render}}(\mathbf{u})\|_2$).

**Teacher distillation.** A frozen teacher provides per-pixel target embeddings $\mathbf{F}_{\text{teacher}}(\mathbf{u})$ (denoted as $\mathbf{z}_{v,\mathbf{u}}$ in Sec. 4.1) after resolution alignment. Importantly, we keep the rasterizer unchanged (standard Euclidean $\alpha$-compositing); our VFM introduces a manifold-aware robust *distillation objective* on $\mathbb{S}^{D-1}$, rather than modifying the fusion mechanism in the renderer.

## 4. Method

### 4.1. Problem Formulation and Notation

We represent a 3D scene as $N$ Gaussian primitives $\mathcal{G} = \{g_i\}_{i=1}^N$ with $g_i = (\mathbf{x}_i, \boldsymbol{\Sigma}_i, \alpha_i, \boldsymbol{\theta}_i, \mathbf{f}_i)$, where $\mathbf{x}_i \in \mathbb{R}^3$, $\boldsymbol{\Sigma}_i \in \mathbb{R}^{3\times3}$, $\alpha_i \in (0,1)$, and $\boldsymbol{\theta}_i$ are standard 3DGS parameters.

**Semantic representation.** Each Gaussian stores a low-dimensional semantic latent $\mathbf{f}_i \in \mathbb{R}^d$ and a lightweight decoder maps it to the teacher feature space: $\mathbf{h}_i = \text{Dec}(\mathbf{f}_i) \in \mathbb{R}^D$. Given a viewpoint $v$ and pixel $\mathbf{u} \in \Omega$, the teacher provides a target embedding $\mathbf{z}_{v,\mathbf{u}} \in \mathbb{R}^D$. The rasterizer renders a pixel embedding $\hat{\mathbf{z}}_{v,\mathbf{u}} \in \mathbb{R}^D$ by standard front-to-back $\alpha$-compositing of $\{\mathbf{h}_i\}$ (Sec. 3). We apply $\ell_2$ normalization to teacher/rendered pixel embeddings before computing hyperspherical (geodesic) losses. *Implementation note:* we keep the rasterizer unchanged; hyperspherical geometry is used only in the distillation objective.

**Rasterization statistics.** The rasterizer also outputs the accumulated opacity $A_{v,\mathbf{u}}$ and depth moments $D_{v,\mathbf{u}}^{(1)}, D_{v,\mathbf{u}}^{(2)}$ (with the same compositing weights). We compute the expected depth $\bar{D}_{v,\mathbf{u}} = \frac{D_{v,\mathbf{u}}^{(1)}}{A_{v,\mathbf{u}}+\epsilon}$ and ray variance $\text{Var}_{v,\mathbf{u}} = \big[\frac{D_{v,\mathbf{u}}^{(2)}}{A_{v,\mathbf{u}}+\epsilon} - \bar{D}_{v,\mathbf{u}}^2\big]_+$, used by VCD (App. A.1).

**Geodesic distance on the hypersphere.** For $\mathbf{a}, \mathbf{b} \in \mathbb{S}^{D-1}$, we use the geodesic distance

$$d(\mathbf{a}, \mathbf{b}) = \arccos\Big(\text{clip}(\langle\mathbf{a}, \mathbf{b}\rangle, -1+\epsilon, 1-\epsilon)\Big). \tag{3}$$

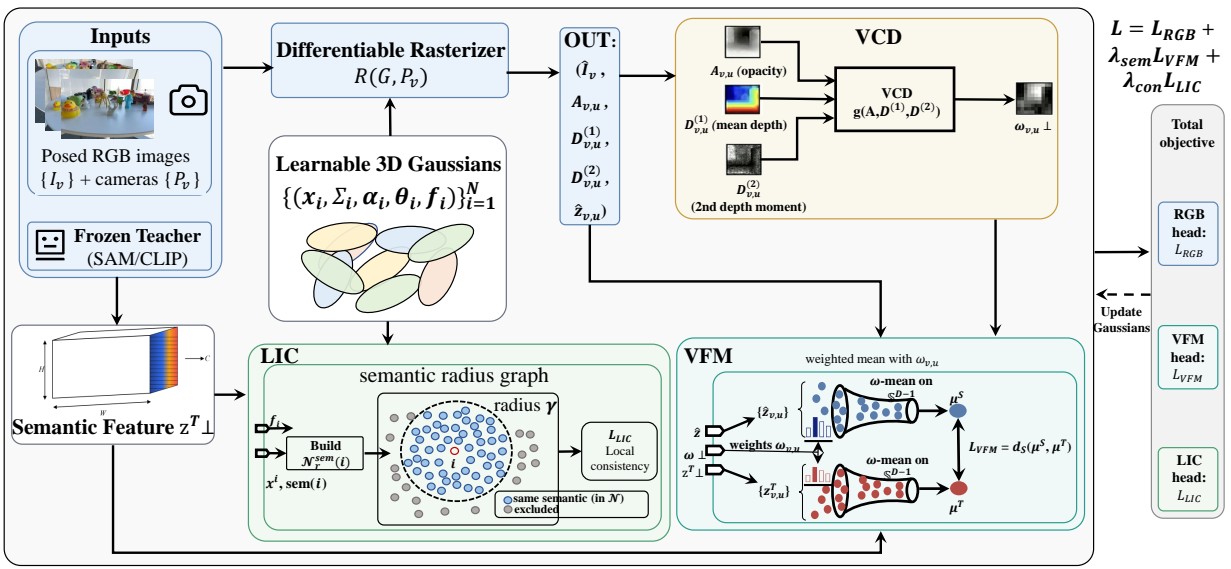

*Figure 2.* **Overview of Rh-3DGS.** Given posed RGB images, a frozen teacher (e.g., SAM/CLIP) provides per-pixel semantic embeddings. Learnable 3D Gaussians are optimized through a differentiable rasterizer, which outputs RGB renders, semantic renders, and rasterization statistics (opacity and depth moments). **VCD** converts these statistics into pixel-wise reliability weights to down-weight occlusion and mixed-depth noise. **VFM** performs visibility-weighted *pixel-wise* hyperspherical distillation, and optionally adds a robust Fréchet-mean stabilizer to improve cross-view consistency and avoid Euclidean feature shrinkage. **LIC** builds a semantic radius graph over Gaussians and applies lightweight contrastive consistency to sharpen local boundaries. The final objective is $L = L_{\text{RGB}} + \lambda_{\text{sem}} L_{\text{VFM}} + \lambda_{\text{con}} L_{\text{LIC}}$.

## 4.2. Overview

Figure 2 overviews the training pipeline of **Rh-3DGS**. Given posed RGB images, we learn a 3D Gaussian scene with an open-vocabulary semantic field. We attach a semantic latent to each Gaussian and use a lightweight decoder to map it to the teacher space ($D \gg d$) during rendering.

A main difficulty is noisy supervision from occlusions and mixed-depth rays. Rh-3DGS addresses it with three modules:

**(1) VCD** computes a stop-gradient pixel reliability weight $w_v(\mathbf{u})$ from rasterization statistics (e.g., accumulated opacity and depth moments) and reweights semantic supervision.

**(2) VFM** performs manifold-consistent distillation on $\mathbb{S}^{D-1}$ to avoid Euclidean feature collapse, and uses robust Fréchet-mean aggregation to reduce residual outliers.

**(3) LIC** applies lightweight neighborhood contrast on Gaussians to enforce local consistency and sharpen boundaries.

We optimize the model end-to-end:

$$\mathcal{L} = \mathcal{L}_{\text{rgb}} + \lambda_{\text{sem}} \mathcal{L}_{\text{VFM}} + \lambda_{\text{con}} \mathcal{L}_{\text{LIC}}, \qquad (4)$$

where $\mathcal{L}_{\text{rgb}}$ is the photometric loss, $\mathcal{L}_{\text{VFM}}$ is the reweighted semantic distillation loss, and $\mathcal{L}_{\text{LIC}}$ regularizes local 3D semantics.

## 4.3. Visibility-Calibrated Distillation (VCD)

Dense distillation from 2D teachers (e.g., CLIP/SAM) (Radford et al., 2021; Kirillov et al., 2023) often contains unreliable supervision (Wang et al., 2025; Tian et al., 2025). The issue is severe near occlusion boundaries and mixed-depth pixels created by $\alpha$-compositing. It causes feature bleeding and view-dependent noise.

We propose **Visibility-Calibrated Distillation (VCD)**. VCD estimates pixel reliability from rasterization statistics of the 3DGS renderer (Kerbl et al., 2023). It outputs a per-pixel weight map and reweights semantic supervision. VCD adds *no extra loss term*. We stop gradients through the weights.

Let $\Omega$ be the image grid and $\mathbf{u} \in \Omega$ be a pixel. For a view $v$, we define the reliability map $\mathcal{W}_v \in \mathbb{R}^{H \times W}$:

$$\mathcal{W}_v(\mathbf{u}) = \mathbf{W}_{op}(\mathbf{u}) \cdot \mathbf{W}_{edge}(\mathbf{u}) \cdot \mathbf{W}_{var}(\mathbf{u}). \qquad (5)$$

Here $\mathcal{N}_q(\cdot)$ is a robust quantile normalization (we use $q = 0.95$). See App. B.2 for $\mathcal{N}_q(\cdot)$.

**Opacity gating.** Low accumulated opacity often indicates weak or unstable contributions. Let $A(\mathbf{u}) \in [0, 1]$ be the accumulated opacity returned by the rasterizer. We use a soft gate:

$$\mathbf{W}_{op}(\mathbf{u}) = \sigma(s \cdot (A(\mathbf{u}) - \tau_{op})), \qquad (6)$$

where $\sigma(\cdot)$ is the sigmoid, $\tau_{op}$ is a threshold, and $s$ controls the transition sharpness.

**Geometric edge suppression.** Teacher embeddings are less reliable near depth discontinuities. We estimate the expected depth from the first raw depth moment:

$$\bar{D}(\mathbf{u}) \doteq \frac{D^{(1)}(\mathbf{u})}{A(\mathbf{u}) + \epsilon}. \tag{7}$$

We suppress pixels with strong gradients on log-depth:

$$\mathbf{W}_{edge}(\mathbf{u}) = \exp\big(-\gamma \cdot \mathcal{N}_q\big(\|\nabla \log(\bar{D}(\mathbf{u}) + \epsilon)\|\big)\big), \tag{8}$$

where $\gamma$ controls the suppression strength.

**Ray variance calibration.** Mixed-depth pixels often have high depth uncertainty. Using $D^{(1)}(\mathbf{u})$ and $D^{(2)}(\mathbf{u})$, we compute

$$\mathbb{E}[z](\mathbf{u}) = \frac{D^{(1)}(\mathbf{u})}{A(\mathbf{u}) + \epsilon}, \qquad \mathbb{E}[z^2](\mathbf{u}) = \frac{D^{(2)}(\mathbf{u})}{A(\mathbf{u}) + \epsilon}, \tag{9}$$

$$\mathrm{Var}(z)(\mathbf{u}) = \Big[\mathbb{E}[z^2](\mathbf{u}) - \mathbb{E}[z](\mathbf{u})^2\Big]_+, \tag{10}$$

and define

$$\mathbf{W}_{var}(\mathbf{u}) = \exp\big(-\beta \cdot \mathcal{N}_q\big(\mathrm{Var}(z)(\mathbf{u})\big)\big), \tag{11}$$

where $\beta$ is a penalty coefficient and $[\cdot]_+$ clamps negative values to 0. We accumulate $A(\mathbf{u})$, $D^{(1)}(\mathbf{u})$, and $D^{(2)}(\mathbf{u})$ with the same compositing weights as rendering. Implementation details are provided in App. A.1.

**Visibility-calibrated reweighting (no extra loss).** Let $\ell_{\mathrm{sem}}(v, \mathbf{u})$ be any per-pixel semantic discrepancy for view $v$. VCD replaces uniform averaging with a weighted average:

$$\mathcal{L}_{\mathrm{sem}}^{\mathrm{cal}}(v) = \frac{\sum_{\mathbf{u} \in \Omega} \mathcal{W}_v(\mathbf{u}) \, \ell_{\mathrm{sem}}(v, \mathbf{u})}{\sum_{\mathbf{u} \in \Omega} \mathcal{W}_v(\mathbf{u}) + \epsilon}. \tag{12}$$

In practice, we use $\mathcal{W}_v \leftarrow \mathrm{stopgrad}(\mathcal{W}_v)$. The model cannot reduce the loss by shrinking weights. $\mathcal{W}_v$ only acts as a reliability mask.

In practice, we instantiate the per-pixel discrepancy as a robust geodesic loss on $\mathbb{S}^{D-1}$:

$$\ell_{\mathrm{sem}}(v, \mathbf{u}) = \rho_\delta\big(d(\tilde{\hat{\mathbf{z}}}_{v,\mathbf{u}}, \tilde{\mathbf{z}}_{v,\mathbf{u}})\big), \tag{13}$$

and obtain the calibrated pixel-wise semantic loss by aggregating over training views:

$$\mathcal{L}_{\mathrm{pix}} = \frac{1}{|\mathcal{V}|} \sum_{v \in \mathcal{V}} \mathcal{L}_{\mathrm{sem}}^{\mathrm{cal}}(v). \tag{14}$$

## 4.4. Visibility-Weighted Fréchet Mean (VFM)

VCD reduces unreliable pixels. However, Euclidean fusion is still problematic. Teacher embeddings are $\ell_2$-normalized and lie on $\mathbb{S}^{D-1}$. Euclidean averaging can shrink norms and drift directions. Some outliers may also remain after VCD. They can bias supervision and hurt view consistency (Radford et al., 2021; Liu et al., 2017; Pennec, 2006). We propose **Visibility-Weighted Fréchet Mean (VFM)**. VFM enforces hyperspherical consistency with an intrinsic geodesic objective on $\mathbb{S}^{D-1}$.

**Manifold-aware distillation vs. manifold rendering.** We *do not* change the 3DGS rasterizer or $\alpha$-compositing. The renderer still forms an *extrinsic* (Euclidean) weighted sum and then normalizes it:

$$\tilde{\mathbf{z}}_{v,\mathbf{u}} = \mathrm{normalize}\left(\sum_k w_k(\mathbf{u}) \, \mathbf{h}_k\right). \tag{15}$$

An intrinsic (Karcher/Fréchet) mean inside the CUDA rasterizer needs per-pixel iterative log/exp updates and trigonometric functions. This adds cost and hurts real-time rendering. Instead, we enforce manifold consistency in the loss. VCD down-weights mixed-depth and occlusion pixels. VFM then aligns directions with a robust geodesic penalty on $\mathbb{S}^{D-1}$.

**Setup.** For view $v$ and pixel $\mathbf{u} \in \Omega$, we use the normalized teacher embedding $\tilde{\mathbf{z}}_{v,\mathbf{u}}$ and the normalized rendered embedding $\tilde{\hat{\mathbf{z}}}_{v,\mathbf{u}}$. VCD provides a reliability weight $w_{v,\mathbf{u}} \geq 0$ (Eq. (5)). We normalize the weights as

$$\bar{w}_{v,\mathbf{u}} = \frac{w_{v,\mathbf{u}}}{\sum_{\mathbf{u} \in \Omega} w_{v,\mathbf{u}} + \epsilon}. \tag{16}$$

**Robust pixel-wise geodesic objective.** We use a Huberized geodesic discrepancy:

$$\ell_{\mathrm{sem}}(v, \mathbf{u}) = \rho_\delta\Big(d\big(\tilde{\hat{\mathbf{z}}}_{v,\mathbf{u}}, \tilde{\mathbf{z}}_{v,\mathbf{u}}\big)\Big), \tag{17}$$

where $d(\cdot, \cdot)$ is the geodesic distance in Eq. (3) and $\rho_\delta(\cdot)$ is the Huber penalty:

$$\rho_\delta(d) = \begin{cases} \frac{1}{2}d^2, & d \leq \delta, \\ \delta\left(d - \frac{1}{2}\delta\right), & d > \delta. \end{cases} \tag{18}$$

We apply VCD reweighting (Eq. (12)) and average over views to obtain $\mathcal{L}_{\mathrm{pix}}$ (Eq. (14)).

**Mean-to-mean stabilizer.** We compute view-wise robust Fréchet means for teacher and rendered features, $\mu_v^t = \arg\min_\mu \sum_{\mathbf{u}} \bar{w}_{v,\mathbf{u}} \rho_\delta(d(\tilde{\mathbf{z}}_{v,\mathbf{u}}, \mu))$ and $\mu_v^s = \arg\min_\mu \sum_{\mathbf{u}} \bar{w}_{v,\mathbf{u}} \rho_\delta(d(\tilde{\hat{\mathbf{z}}}_{v,\mathbf{u}}, \mu))$, and align them by

$$\mathcal{L}_{\mathrm{mean}} = \frac{1}{|\mathcal{V}|} \sum_{v \in \mathcal{V}} d\big(\mu_v^s, \mu_v^t\big), \tag{19}$$

where the mean solver is given in App. B.3.

**Final VFM objective.**

$$\mathcal{L}_{\text{VFM}} = \mathcal{L}_{\text{pix}} + \lambda_{\text{mean}}\mathcal{L}_{\text{mean}}. \qquad (20)$$

We can anneal $\delta$ during training. A larger $\delta$ gives coarse alignment early. A smaller $\delta$ refines semantics later.

### 4.5. Lightweight Consistency Contrast (LIC)

VCD and VFM provide robust *pixel-to-teacher* supervision, yet the learned 3D semantic field can remain locally inconsistent, especially near object boundaries and sparsely observed regions. We introduce **Lightweight Consistency Contrast (LIC)**, a simple contrastive regularizer directly applied to Gaussian semantics.

**Setup and similarity.** Each Gaussian $g_i$ has center $\mathbf{x}_i \in \mathbb{R}^3$ and a decoded teacher-space feature $\mathbf{h}_i = \text{Dec}(\mathbf{f}_i) \in \mathbb{R}^D$. We use the normalized embedding $\tilde{\mathbf{h}}_i = \mathbf{h}_i/\|\mathbf{h}_i\|_2 \in \mathbb{S}^{D-1}$. At each LIC step, we sample a mini-set of Gaussians $\mathcal{I}$ (boundary-biased sampling for efficiency; see App. B.4). Cosine similarity with temperature $T$ is

$$s_{ab} = \frac{\langle \tilde{\mathbf{h}}_{i_a}, \tilde{\mathbf{h}}_{i_b} \rangle}{T}. \qquad (21)$$

**Positives via local pseudo-instances.** Within the sampled set, we build local pseudo-instance IDs $\pi(i) \in \{1, \dots\}$ by clustering teacher-aligned features in a spatial neighborhood (details in App. B.4). For an anchor $i_a \in \mathcal{I}$, we define positives as nearby Gaussians that share the same pseudo-instance:

$$\mathcal{I}_\pi(i_a) = \{ i \in \mathcal{I} \mid \pi(i) = \pi(i_a) \}, \qquad (22)$$

$$\mathcal{P}(i_a) = \{ i_b \in \mathcal{I}_\pi(i_a) \setminus \{i_a\} \mid \|\mathbf{x}_{i_a} - \mathbf{x}_{i_b}\|_2 < r \}. \qquad (23)$$

All other in-batch samples are negatives:

$$\mathcal{N}(i_a) = \mathcal{I} \setminus (\mathcal{P}(i_a) \cup \{i_a\}). \qquad (24)$$

*Remark.* Disabling clustering (i.e., setting all $\pi(i)$ identical) reduces Eq. (23) to a radius-only neighborhood, which we report as an ablation.

**Multi-positive InfoNCE.** We apply a multi-positive InfoNCE loss (Oord et al., 2018; Chen et al., 2020):

$$\mathcal{L}_{\text{LIC}}(i_a) = -\log \frac{\sum\limits_{i_b \in \mathcal{P}(i_a)} \exp(s_{ab})}{\sum\limits_{i_c \in \mathcal{I} \setminus \{i_a\}} \exp(s_{ac})}, \qquad (25)$$

and average over anchors,

$$\mathcal{L}_{\text{LIC}} = \frac{1}{|\mathcal{I}|} \sum_{i_a \in \mathcal{I}} \mathcal{L}_{\text{LIC}}(i_a). \qquad (26)$$

**Efficiency.** LIC is computed intermittently and skips anchors with too few positives for stability. This keeps overhead negligible; full sampling/clustering details are deferred to App. B.4.

## 5. Experiments

We first describe the experimental setup, including datasets, baselines, and evaluation metrics. We then report quantitative and qualitative results to evaluate Rh-3DGS. Finally, we conduct ablation studies to analyze the impact of each component.

### 5.1. Experimental Setup

**Implementation.** We implement all methods in PyTorch and train on a single NVIDIA GeForce RTX 4090 GPU. Unless stated otherwise, we run 30,000 iterations with Adam, following the standard 3DGS optimization protocol (Kerbl et al., 2023). We optimize RGB reconstruction and add our semantic objectives with fixed weights. Full hyperparameters and schedules are deferred to App. A.2.

**Datasets.** We evaluate Rh-3DGS on three benchmarks: (i) **LERF** (Kerr et al., 2023), multi-view scenes with mask-based open-vocabulary queries; (ii) **3D-OVS** (Liu et al., 2023), a standardized benchmark for open-vocabulary 3D segmentation; (iii) **ScanNet** (Dai et al., 2017), a large-scale indoor RGB-D dataset widely used for 3D semantic segmentation.

**Baselines.** We compare against Gaussian-based baselines: **Feature 3DGS** (Zhou et al., 2024), **LangSplat** (Qin et al., 2024), **OpenGaussian** (Wu et al., 2024), **LaGa** (Cen et al., 2025), **InstanceGaussian** (Li et al., 2025a), **ILGS** (Jang & Kim, 2025), **LangSplatV2** (Li et al., 2025b) and **VALA** (Wang et al., 2025).

**Metrics.** We follow official evaluation protocols. For **LERF** and **3D-OVS**, we report **mIoU** and **mBIoU** (IoU computed on boundary bands around contours). For **ScanNet**, we report **mIoU** and **mAcc** (mean per-class accuracy).

### 5.2. Quantitative Results

**LERF Dataset.** Tab. 2 reports open-vocabulary segmentation on LERF using mIoU and mBIoU. Tab. 3 reports rendering quality using PSNR, SSIM, and LPIPS. Rh-3DGS achieves the best results on both tables. Compared with the strongest baseline, Rh-3DGS improves mIoU from 76.07 to 82.07 and mBIoU from 55.45 to 67.66. The gain on mBIoU is especially large, which indicates sharper and more accurate boundaries. We also observe consistent improvements in rendering metrics. Rh-3DGS reaches the highest PSNR and SSIM, and matches the best LPIPS. These results sug-

gest that our semantic training does not hurt radiance-field reconstruction. The improvements are most visible on thin structures and boundary transitions. VCD down-weights pixels affected by occlusion and mixed-depth rays. VFM aggregates embeddings on the hypersphere and avoids feature collapse. This combination yields cleaner masks and more view-consistent rendering.

*Table 2.* Quantitative mIoU(%) and mBIoU(%) results on the LERF dataset.

| Method | mIoU(%) ↑ | mBIoU(%) ↑ |
|---|---|---|
| Feature 3DGS[CVPR'24] | 29.34 | 24.23 |
| LangSplat[CVPR'24] | 29.99 | 24.02 |
| OpenGaussian[NeurIPS'24] | 43.77 | 37.27 |
| LaGa[ICML'25] | 63.24 | 55.45 |
| InstanceGaussian[CVPR'25] | 42.43 | 35.99 |
| ILGS[ICCV'25] | 76.07 | 51.83 |
| LangSplatV2[NeurIPS'25] | 59.62 | 47.93 |
| VALA[arxiv'25] | 61.78 | 50.72 |
| Rh-3DGS (Ours) | **82.07** | **67.66** |

*Table 3.* Quantitative PSNR, SSIM and LPIPS results on the LERF dataset.

| Method | PSNR ↑ | SSIM ↑ | LPIPS ↓ |
|---|---|---|---|
| Feature 3DGS | 29.06 | 0.90 | 0.16 |
| LangSplat | 25.27 | 0.84 | 0.23 |
| OpenGaussian | 27.10 | 0.88 | 0.21 |
| LaGa | 28.58 | 0.89 | 0.17 |
| InstanceGaussian | 29.06 | 0.90 | **0.13** |
| ILGS | 27.49 | 0.87 | 0.21 |
| Rh-3DGS (Ours) | **29.49** | **0.92** | **0.13** |

**3D-OVS Dataset.** Tab. 4 reports results on 3D-OVS using mIoU and mBIoU. Rh-3DGS again achieves the best performance. It improves mIoU from 95.13 to 96.88 and mBIoU from 86.02 to 89.62 over the strongest baseline. The consistent gain on mBIoU indicates better boundary fidelity under open-vocabulary queries. This dataset contains frequent occlusions and depth-layer mixing. In such cases, naive distillation can introduce view-dependent noise. VCD reduces these corrupted signals via visibility-aware weighting. VFM further stabilizes fusion by using a robust Fréchet mean on $\mathbb{S}^{D-1}$. Together they improve cross-view semantic consistency.

**ScanNet Dataset.** Tab. 5 reports ScanNet semantic segmentation using mIoU and mAcc under 19/15/10-class settings. Rh-3DGS achieves the best results across all splits. For the 19-class setting, it improves mIoU/mAcc from 36.2/51.9 to 40.8/59.0. Similar gains hold for the

*Table 4.* Quantitative mIoU(%) and mBIoU(%) results on the 3D-OVS dataset.

| Method | mIoU(%) ↑ | mBIoU(%) ↑ |
|---|---|---|
| Feature 3DGS[CVPR'24] | 86.51 | 80.83 |
| LangSplat[CVPR'24] | 93.43 | 85.33 |
| OpenGaussian[NeurIPS'24] | 91.08 | 82.79 |
| LaGa[ICML'25] | 95.13 | 86.02 |
| InstanceGaussian[CVPR'25] | 86.86 | 80.11 |
| ILGS[ICCV'25] | 93.76 | 81.09 |
| LangSplatV2[NeurIPS'25] | 94.58 | 86.14 |
| VALA[arxiv'25] | 95.11 | 86.93 |
| Rh-3DGS (Ours) | **96.88** | **89.62** |

*Table 5.* Quantitative mIoU(%) and mAcc(%) results on the ScanNet dataset.

| Method | Classes 19 mIoU / mAcc | Classes 15 mIoU / mAcc | Classes 10 mIoU / mAcc |
|---|---|---|---|
| Feature3DGS | 10.7 / 20.8 | 18.3 / 29.7 | 25.8 / 47.7 |
| LangSplat | 3.9 / 10.2 | 5.3 / 15.4 | 9.1 / 24.9 |
| OpenGaussian | 22.1 / 37.9 | 27.4 / 45.3 | 35.2 / 50.9 |
| LaGa | 32.5 / 49.1 | 35.8 / 53.5 | 42.6 / 63.5 |
| InstanceGaussian | 36.2 / 51.9 | 38.5 / 54.4 | 48.0 / 65.2 |
| LangSplatV2 | 25.9 / 46.2 | 34.0 / 50.0 | 40.3 / 61.8 |
| VALA | 32.1 / 50.1 | 35.1 / 54.8 | 46.2 / 65.6 |
| ILGS | 36.8 / 51.8 | 39.0 / 58.2 | 48.5 / 66.3 |
| Rh-3DGS | **40.8 / 59.0** | **42.4 / 63.7** | **52.4 / 69.0** |

15-class and 10-class splits. These results show that our semantic field learning generalizes beyond open-vocabulary benchmarks. It also improves region-level recognition and per-class reliability in closed-set evaluation.

### 5.3. Qualitative Results.

Fig. 3 shows visual comparisons on LERF under open-vocabulary queries. This setting is hard because objects are small, thin, and often partially occluded. Baseline methods often produce boundary bleeding and scattered activations. Rh-3DGS outputs tighter masks with cleaner separation. It better preserves thin structures and sharp edges (e.g., "egg" and "coffee mug"). It also reduces background leakage in cluttered scenes (e.g., "rubber duck with red hat" and "red apple"). Across views, our predictions look more stable and show fewer view-dependent artifacts.

Fig. 4 shows qualitative results on ScanNet in cluttered indoor scenes. Rh-3DGS localizes target regions on the reconstructed mesh with clear boundaries. It handles heavy occlusion and close-by categories well (e.g., "bed/sofa/table"

and "counter/door/floor"). These examples indicate that our 3D semantic field remains consistent in complex indoor layouts.

More qualitative results on LERF and 3D-OVS, including multi-view consistency visualizations, are provided in App. C.

### 5.4. Ablation Study

We ablate each component on LERF (*figurines*). All variants use the same training schedule, teacher, resolution, and evaluation protocol.

**Results.** Table 6 shows module-level ablations. The full model (VCD+VFM+LIC) achieves the best performance (81.62 mIoU, 58.11 mBIoU). Adding VCD to VFM+LIC improves by +5.38 mIoU and +6.41 mBIoU. Adding LIC to VCD+VFM improves by +6.54 mIoU and +5.61 mBIoU. Gains are larger on mBIoU, indicating sharper boundaries under occlusion and mixed-depth rays.

**Why VCD alone is weaker.** VCD alone improves the baseline (65.37 vs. 60.56 mIoU), but it is not sufficient. It filters unreliable pixels and reduces effective supervision. VFM and LIC complement VCD: VFM enforces hyperspherical consistency on $\mathbb{S}^{D-1}$, and LIC strengthens local 3D boundary coherence.

**Efficiency.** We benchmark **inference rendering** (RGB + semantic) on LERF *figurines* at $1280 \times 720$ on an RTX 4090. As in Tab. 6, the full model runs at 301.64 FPS with 3.23 GB peak memory. Compared to the baseline (328.59 FPS, 2.65 GB), all components add only 8.2% FPS drop and 0.58 GB memory, while improving mIoU by +21.06 and mBIoU by +26.35.

**Decomposing VFM.** VFM is defined as $L_{\mathrm{VFM}} = L_{\mathrm{pix}} + \lambda_{\mathrm{mean}} L_{\mathrm{mean}}$, where $L_{\mathrm{pix}}$ is the pixel-wise Huberized geodesic supervision and $L_{\mathrm{mean}}$ is the view-wise robust Fréchet-mean stabilizer. To isolate their roles more clearly, we disable VCD and LIC and compare the baseline, $L_{\mathrm{pix}}$ only, $L_{\mathrm{mean}}$ only, and the full VFM objective. As shown in Tab. 7, $L_{\mathrm{pix}}$ alone already yields a substantial improvement over the baseline, while $L_{\mathrm{mean}}$ alone is also beneficial but smaller. Their combination performs best. This indicates that the main manifold-aware gain already appears at the pixel level, while the Fréchet-mean term further stabilizes cross-view aggregation.

**VCD dynamics over training.** We further analyze whether VCD provides stable reliability calibration throughout optimization. As shown in Tab. 8, the average reliability weight increases from 0.52 at 1k iterations to about 0.61 after 8k iterations and then remains stable. The fraction of

*Table 6.* **Ablation on LERF (figurines).**

| Cfg (VCD/VFM/LIC) | mIoU | mBIoU | FPS | Mem.(GB) |
|---|---|---|---|---|
| ✗ ✗ ✗ | 60.56 | 31.76 | 328.59 | **2.65** |
| ✗ ✗ ✓ | 75.70 | 44.83 | 358.57 | 2.67 |
| ✗ ✓ ✗ | 73.76 | 44.68 | 274.26 | 2.68 |
| ✓ ✗ ✗ | 65.37 | 37.07 | **389.57** | 2.68 |
| ✗ ✓ ✓ | 76.24 | 51.70 | 282.46 | 2.67 |
| ✓ ✗ ✓ | 66.97 | 44.70 | 319.23 | **2.65** |
| ✓ ✓ ✗ | 75.08 | 52.50 | 282.10 | 2.68 |
| ✓ ✓ ✓ | **81.62** | **58.11** | 301.64 | 3.23 |

*Table 7.* **Decomposition of VFM on LERF (figurines).** VCD and LIC are disabled. $L_{\mathrm{pix}}$ provides the main gain, $L_{\mathrm{mean}}$ alone is beneficial but smaller, and the full VFM objective performs best.

| Setting | $L_{\mathrm{pix}}$ | $L_{\mathrm{mean}}$ | mIoU | mBIoU |
|---|---|---|---|---|
| Baseline | ✗ | ✗ | 60.56 | 31.76 |
| + $L_{\mathrm{pix}}$ | ✓ | ✗ | 71.81 | 42.94 |
| + $L_{\mathrm{mean}}$ | ✗ | ✓ | 66.46 | 36.76 |
| + full VFM | ✓ | ✓ | **73.76** | **44.68** |

*Table 8.* **Training-time dynamics of VCD reliability weights on LERF (figurines).** Boundary pixels consistently receive lower weights than non-boundary pixels, showing that VCD remains effective throughout optimization.

| Iter | Mean $W$ | Pixels $W < 0.1$ | Boundary mean $W$ | Non-boundary mean $W$ |
|---|---|---|---|---|
| 1k | 0.52 | 0.16 | 0.10 | 0.62 |
| 8k | 0.60 | 0.14 | 0.13 | 0.71 |
| 20k | 0.61 | 0.13 | 0.14 | 0.72 |
| 30k | 0.61 | 0.13 | 0.14 | 0.73 |

strongly down-weighted pixels decreases only moderately from 0.16 to 0.13. More importantly, boundary pixels consistently receive much lower weights than non-boundary pixels, e.g., 0.10 vs. 0.62 at 1k and 0.14 vs. 0.73 at 30k. This shows that VCD persistently suppresses unreliable supervision near occlusion and mixed-depth regions, rather than only becoming calibrated after convergence.

**LIC scheduling stability.** We also study two scheduling choices of LIC: the warm-up start iteration and the pseudo-instance refresh interval. As shown in Tab. 9, activating LIC from the beginning is less effective because pseudo-instances are unstable in the early stage. Starting LIC at 3k iterations gives the best result. For pseudo-instance refresh, 500 iterations achieves the best trade-off between adaptation and overhead. More frequent refresh brings higher cost, while overly sparse refresh weakens adaptation.

### 6. Conclusion

We present **Rh-3DGS** for robust open-vocabulary 3D semantics in 3D Gaussian Splatting. Dense 2D teacher supervision is often unreliable in 3DGS, especially near occlu-

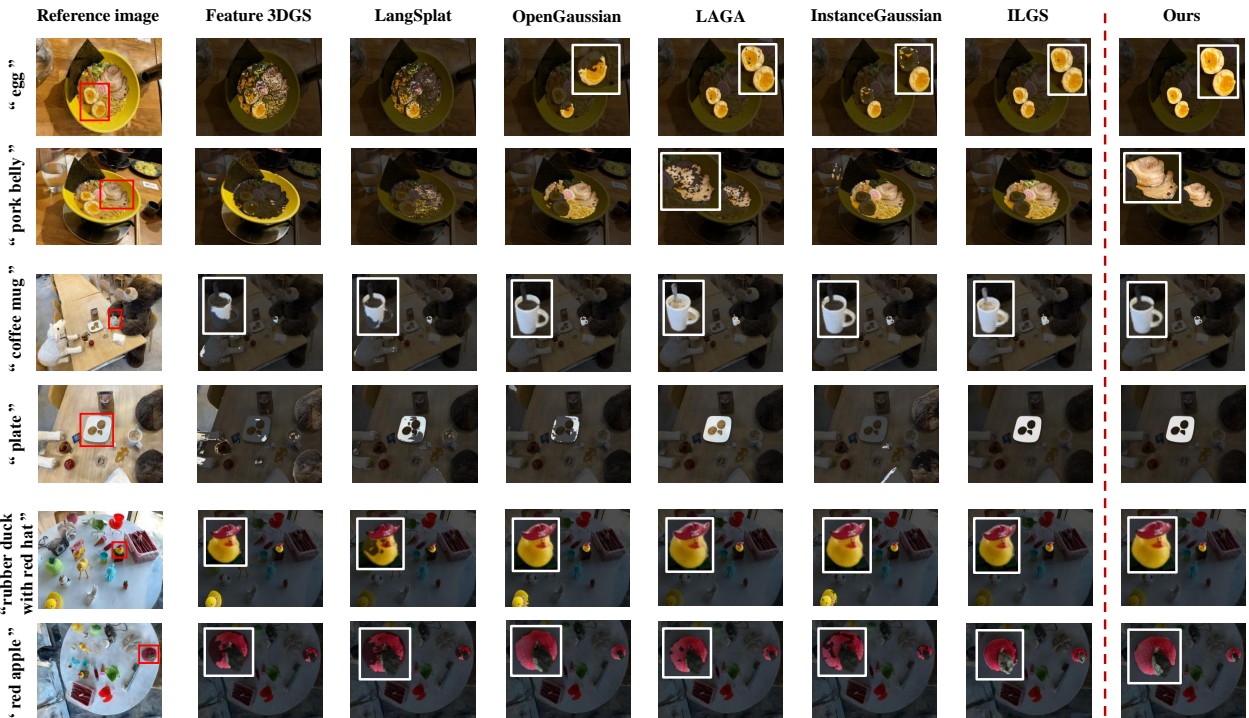

*Figure 3.* **Open-vocabulary inference and evaluation with Rh-3DGS.** Given a text prompt, we compute pixel-wise similarity between the CLIP text embedding and the rendered semantic map to obtain $S_v(u)$. For 2D evaluation, we threshold $S_v$ to produce masks and compute mIoU. Optionally, we aggregate $S_v$ to a Gaussian-level score to obtain a 3D mask for editing demos.

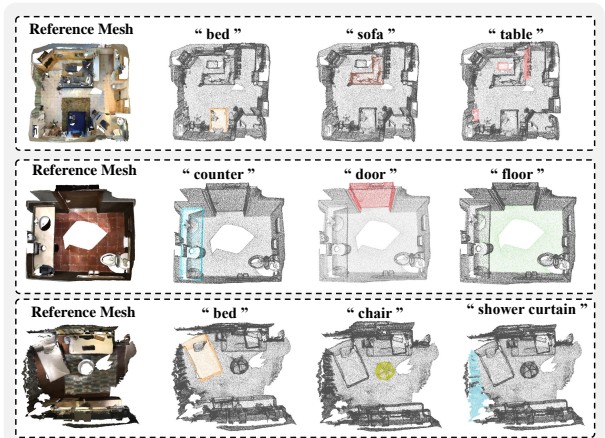

*Figure 4.* **Qualitative results on ScanNet.** We visualize open-vocabulary queries on reconstructed scenes. Left: reference mesh. Right: Rh-3DGS predictions for different text queries (e.g., "bed", "sofa", "table", "counter", "door", "floor", "chair", and "shower curtain"). Each row shows a different scene. Rh-3DGS localizes semantic regions with clean boundaries under clutter and occlusion.

*Table 9.* **LIC scheduling analysis on LERF (figurines).** The default setting, 3k warm-up and 500-iteration refresh, gives the best trade-off.

| Factor | Setting | mIoU | Observation |
|---|---|---|---|
| LIC warm-up start | 0 | 80.94 | too early; pseudo-instances less stable |
| LIC warm-up start | 1k | 81.28 | improved, but still slightly noisy |
| LIC warm-up start | 3k | **81.62** | best overall |
| LIC warm-up start | 5k | 81.31 | slightly delayed regularization |
| Cluster refresh interval | 100 iters | 81.39 | more frequent; higher overhead |
| Cluster refresh interval | 500 iters | **81.62** | best trade-off |
| Cluster refresh interval | 1000 iters | 81.17 | too sparse; weaker adaptation |

stop-gradient. It adds no extra loss term. **VFM** aggregates embeddings on $\mathbb{S}^{D-1}$ to avoid Euclidean feature collapse and reduce mixing. **LIC** regularizes the 3D semantic field with local contrast for sharper boundaries. Experiments on LERF, 3D-OVS, and ScanNet show consistent improvements, especially on boundary metrics. Ablations confirm that the modules are complementary.

**Limitations and Future Work.** Rh-3DGS inherits teacher biases and adds training overhead from extra statistics and neighborhood queries. Future work will extend to dynamic scenes, multi-teacher distillation, and more efficient implementations.

sions and mixed-depth rays. This causes feature bleeding and view-dependent semantics. Rh-3DGS combines three components. **VCD** computes a pixel-wise reliability map from rasterization statistics and reweights distillation with

## Impact Statement

This paper presents work whose goal is to advance the field of Machine Learning. There are many potential societal consequences of our work, none which we feel must be specifically highlighted here.

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

# Appendix Contents

# A. Implementation Details and Reproducibility

## A.1. Rasterizer Outputs for VCD

VCD uses three per-pixel statistics from the 3DGS rasterizer: accumulated opacity $A(\mathbf{u})$ and raw depth moments $\mathbf{D}^{(1)}(\mathbf{u})$ and $\mathbf{D}^{(2)}(\mathbf{u})$. They are accumulated with the same front-to-back $\alpha$-compositing weights as color/feature rendering.

Let $\mathcal{N}(\mathbf{u})$ be the ordered set of Gaussians contributing to pixel $\mathbf{u}$. For the $k$-th contributor, let $\alpha'_k(\mathbf{u})$ be its effective opacity and $T_k(\mathbf{u})$ be the transmittance. We define

$$A(\mathbf{u}) \;=\; \sum_k T_k(\mathbf{u})\, \alpha'_k(\mathbf{u}). \tag{27}$$

Let $z_k$ be the depth of the $k$-th Gaussian center in camera coordinates (after projection). We accumulate the first and second raw depth moments as

$$\mathbf{D}^{(1)}(\mathbf{u}) = \sum_k T_k(\mathbf{u})\, \alpha'_k(\mathbf{u})\, z_k,$$
$$\mathbf{D}^{(2)}(\mathbf{u}) = \sum_k T_k(\mathbf{u})\, \alpha'_k(\mathbf{u})\, z_k^2. \tag{28}$$

**Implementation.**    We obtain $\mathbf{D}^{(1)}$ with the standard renderer by accumulating one extra scalar per pixel. To obtain $\mathbf{D}^{(2)}$, we add a lightweight accumulation buffer for $z_k^2$ with the same weights. This change does not affect rendering or gradients of the original RGB loss. It only provides additional statistics for VCD weight computation.

## A.2. Training Details

**Hardware.**    All experiments are trained on a single NVIDIA GeForce RTX 4090 GPU.

**Optimizer and iterations.**    We train for 30,000 iterations using Adam. We follow the standard 3DGS learning-rate protocol (Kerbl et al., 2023). The position learning rate decays from $1.6 \times 10^{-4}$ to $1.6 \times 10^{-6}$. We use fixed learning rates for other parameters: spherical-harmonic coefficients $2.5 \times 10^{-3}$, opacity $5 \times 10^{-2}$, scaling $5 \times 10^{-3}$, and rotation $1 \times 10^{-3}$. The semantic feature learning rate is $2.5 \times 10^{-3}$.

**Photometric loss.**    We use an RGB reconstruction loss that combines $\mathcal{L}_1$ and D-SSIM. We set $\lambda_{\mathrm{dssim}} = 0.2$.

**Densification and opacity reset.**    We densify every 100 iterations from iteration 500 to 15,000. We use a gradient threshold of $2 \times 10^{-4}$. We cap the number of Gaussians at 500,000. We reset opacity every 3,000 iterations.

**Loss weights.**    We set $\lambda_{\mathrm{rgb}} = 1.0$ and $\lambda_{\mathrm{sem}} = 1.0$. We set $\lambda_{\mathrm{VFM}} = 0.05$ and $\lambda_{\mathrm{LIC}} = 0.02$.

**VCD hyperparameters.**    We use opacity threshold $\tau_{op} = 0.3$ and sharpness $s = 10$. We set the edge sensitivity to $\gamma = 3.0$. We set the variance penalty to $\beta = 2.0$. We use quantile normalization with $q = 0.95$.

**VFM hyperparameters.**    We use a Huber-style robust loss with a scheduled threshold $\delta$. Specifically, we linearly anneal $\delta$ during the first $80\%$ of training iterations and keep it fixed for the remaining $20\%$:

$$\delta(t) = \delta_{\mathrm{max}} + (\delta_{\mathrm{min}} - \delta_{\mathrm{max}}) \cdot \min\left(1, \frac{t}{T_\delta}\right), \tag{29}$$

where $t$ is the iteration index, $T_\delta = 0.8\, T_{\mathrm{train}}$, and we set $(\delta_{\mathrm{max}}, \delta_{\mathrm{min}}) = (0.2, 0.05)$.

## A.3. Teacher Models and Feature Extraction

We use two frozen 2D teachers for semantic supervision. All teacher parameters are fixed during training.

**SAM teacher.**    We use Segment Anything Model (SAM) with the ViT-H backbone. We load the official checkpoint `sam_vit_h_4b8939`. We use SAM to extract dense image features (and optional mask-related features when needed).

**OpenCLIP teacher.**    We use OpenCLIP as the vision–language teacher. We adopt `ViT-B-16` pretrained on `laion2b_s34b_b88k`. We use its image encoder to produce $\ell_2$-normalized embeddings for open-vocabulary queries.

## A.4. LIC Scheduling and Efficiency

**Delayed activation.**    LIC is activated after iteration 3,000 rather than from the beginning. This warm-up allows the 3D Gaussian field to stabilize geometrically and learn meaningful semantics through VFM distillation before applying contrastive regularization. Empirically, enabling LIC too early can destabilize training because initial features are noisy and yield unreliable positive/negative relations.

**Mini-batch sampling.**    At each LIC step, we sample a mini-batch of $B = 512$ Gaussians from the full set $\mathcal{G}$ ($N \approx 200\mathrm{k}$–500k). To balance locality and coverage, we adopt a *patch-based sampling* strategy: we first select $B/P$ seed Gaussians uniformly at random, and then sample $P = 128$ Gaussians from each seed's spatial neighborhood to form patches. This maintains locality within each patch while covering diverse scene regions across patches.

**Instance clustering.**    When instance clustering is enabled, we maintain pseudo-instance assignments via GPU-accelerated KMeans with $K = 20$ clusters, refreshed every 500 iterations, in a standardized joint space of position and semantics. Concretely, we build the clustering input as a weighted concatenation $\mathbf{q}_i = [\sqrt{0.3}\,\mathrm{Norm}(\mathbf{x}_i),\ \sqrt{0.7}\,\mathrm{Norm}(\tilde{\mathbf{h}}_i)]$, where $\mathrm{Norm}(\cdot)$ standardizes each component to zero mean and unit variance. This online clustering is infrequent and lightweight compared to the per-iteration training cost.

**Efficiency.** LIC adds minimal overhead due to: (1) small batch size ($B = 512$) vs. total Gaussians ($N \sim 300k$); (2) no memory bank or momentum encoder; (3) simple pairwise distance and cosine similarity on GPU; (4) sparse evaluation (only when iter $\geq 3000$, and intermittently thereafter). In practice, LIC increases per-iteration time by less than $5\%$ while providing consistent gains in 3D semantic coherence.

**Hyperparameters.** The contrastive temperature is $T = 0.07$. For spatial positives, we use radius $r = 0.4$ (normalized scene coordinates). Hard negatives within $\eta r$ receive a logit bias of $\log \gamma$ (default $\gamma = 1.0$, i.e., no bias). Anchors with fewer than $m = 1$ positive are excluded from the loss to avoid degenerate gradients.

## B. Mathematical Details

### B.1. Riemannian Preliminaries on the Hypersphere

We model semantic embeddings as points on the unit hypersphere $\mathbb{S}^{D-1} = \{\mathbf{x} \in \mathbb{R}^D \mid \|\mathbf{x}\|_2 = 1\}$. For $\mathbf{a}, \mathbf{b} \in \mathbb{S}^{D-1}$, the geodesic distance is

$$d_{\mathbb{S}}(\mathbf{a}, \mathbf{b}) = \arccos(\langle \mathbf{a}, \mathbf{b} \rangle), \tag{30}$$

where $\langle \cdot, \cdot \rangle$ denotes the Euclidean inner product. The tangent space at $\boldsymbol{\mu} \in \mathbb{S}^{D-1}$ is

$$T_{\boldsymbol{\mu}} \mathbb{S}^{D-1} = \{\mathbf{v} \in \mathbb{R}^D \mid \langle \mathbf{v}, \boldsymbol{\mu} \rangle = 0\}. \tag{31}$$

**Log map.** Let $\theta = \arccos(\langle \boldsymbol{\mu}, \mathbf{x} \rangle) \in [0, \pi]$. The logarithmic map from $\mathbf{x} \in \mathbb{S}^{D-1}$ to $T_{\boldsymbol{\mu}} \mathbb{S}^{D-1}$ is

$$\log_{\boldsymbol{\mu}}(\mathbf{x}) = \frac{\theta}{\sin \theta + \epsilon} \left( \mathbf{x} - \cos \theta \, \boldsymbol{\mu} \right), \tag{32}$$

where $\epsilon > 0$ is a small constant for numerical stability. Note that $\log_{\boldsymbol{\mu}}(\mathbf{x}) \in T_{\boldsymbol{\mu}} \mathbb{S}^{D-1}$ by construction.

**Retraction.** To map a tangent vector back to the sphere, we use a simple retraction (first-order approximation to the exponential map):

$$\mathcal{R}_{\boldsymbol{\mu}}(\mathbf{v}) = \frac{\boldsymbol{\mu} + \mathbf{v}}{\|\boldsymbol{\mu} + \mathbf{v}\|_2}. \tag{33}$$

**Why Euclidean averaging is inconsistent on $\mathbb{S}^{D-1}$.** Let $\{\mathbf{x}_i\}_{i=1}^n \subset \mathbb{S}^{D-1}$ and consider their Euclidean average $\mathbf{m} = \frac{1}{n} \sum_i \mathbf{x}_i$. By the triangle inequality,

$$\|\mathbf{m}\|_2 = \left\| \frac{1}{n} \sum_{i=1}^n \mathbf{x}_i \right\|_2 \leq \frac{1}{n} \sum_{i=1}^n \|\mathbf{x}_i\|_2 = 1, \tag{34}$$

with equality iff all $\mathbf{x}_i$ are identical (or perfectly aligned). Hence, Euclidean fusion of spherical features induces *norm shrinkage*. Renormalizing $\mathbf{m}$ back to the sphere changes the direction nonlinearly and can distort angular relationships, motivating manifold-consistent aggregation (our VFM).

### B.2. VCD: From Rasterization Statistics to Reliability Weights

VCD assigns a reliability weight $\mathcal{W}(\mathbf{u})$ to each pixel $\mathbf{u} \in \Omega$. It uses three cues from the differentiable 3DGS rasterizer: accumulated opacity, depth edges, and ray depth variance.

**Rasterizer quantities (opacity and raw moments).** For pixel $\mathbf{u}$, the rasterizer composites projected Gaussians in front-to-back order. For the $k$-th contributor, we define the compositing weight

$$w_k(\mathbf{u}) \doteq T_k(\mathbf{u}) \, \alpha'_k(\mathbf{u}), \tag{35}$$

where $\alpha'_k(\mathbf{u}) \in [0, 1]$ is the effective per-pixel opacity contribution and $T_k(\mathbf{u})$ is the transmittance before this contribution. The accumulated opacity is

$$A(\mathbf{u}) \doteq \sum_k w_k(\mathbf{u}) = 1 - T_{\text{end}}(\mathbf{u}). \tag{36}$$

We also accumulate the raw first/second depth moments:

$$D^{(1)}(\mathbf{u}) \doteq \sum_k w_k(\mathbf{u}) z_k, \qquad D^{(2)}(\mathbf{u}) \doteq \sum_k w_k(\mathbf{u}) z_k^2, \tag{37}$$

where $z_k$ is the depth of the $k$-th contributor (in camera space). In our code, these correspond to `render_opacity`, `render_depth`, and `render_depth2`.

**Robust quantile normalization.** Given a nonnegative map $\mathbf{M}$, we normalize it by the per-image $q$-quantile:

$$\mathcal{N}_q(\mathbf{M})(\mathbf{u}) \doteq \text{clip}\left(\frac{\mathbf{M}(\mathbf{u})}{Q_q(\mathbf{M}) + \epsilon}, 0, 1\right), \tag{38}$$

where $Q_q(\mathbf{M})$ is computed over valid pixels of the image. We use $q = 0.95$ in all experiments.

**(i) Opacity gating.** We softly suppress pixels with low accumulated opacity:

$$\mathbf{W}_{op}(\mathbf{u}) = \sigma(s \cdot (A(\mathbf{u}) - \tau_{op})). \tag{39}$$

**(ii) Edge suppression from expected depth gradients.** Depth discontinuities often indicate occlusion boundaries. We compute the expected depth

$$\bar{D}(\mathbf{u}) \doteq \mathbb{E}[z](\mathbf{u}) = \frac{D^{(1)}(\mathbf{u})}{A(\mathbf{u}) + \epsilon}, \tag{40}$$

and the Sobel gradient magnitude on log-depth:

$$G(\mathbf{u}) \doteq \left\|\nabla \log(\bar{D}(\mathbf{u}) + \epsilon)\right\|_2. \tag{41}$$

We then define

$$\mathbf{W}_{edge}(\mathbf{u}) = \exp(-\gamma \cdot \mathcal{N}_q(G)(\mathbf{u})). \tag{42}$$

**(iii) Ray variance calibration.** Mixed-depth pixels are characterized by high uncertainty along the ray. Using the raw moments, we estimate

$$\mathbb{E}[z^2](\mathbf{u}) = \frac{D^{(2)}(\mathbf{u})}{A(\mathbf{u}) + \epsilon}, \qquad \text{Var}(z)(\mathbf{u}) = \left[\mathbb{E}[z^2](\mathbf{u}) - \mathbb{E}[z](\mathbf{u})^2\right]_+, \tag{43}$$

and define

$$\mathbf{W}_{var}(\mathbf{u}) = \exp(-\beta \cdot \mathcal{N}_q(\text{Var}(z))(\mathbf{u})). \tag{44}$$

**Final reliability map and gradient stopping.** We combine the three terms: $\mathcal{W}(\mathbf{u}) = \mathbf{W}_{op}(\mathbf{u})\,\mathbf{W}_{edge}(\mathbf{u})\,\mathbf{W}_{var}(\mathbf{u})$. We set $\mathcal{W} \leftarrow \text{stopgrad}(\mathcal{W})$ during training. This avoids degenerate solutions that reduce the loss by shrinking weights.

### B.3. VFM: Huberized Visibility-Weighted Fréchet Mean on $\mathbb{S}^{D-1}$

This section describes the solver for the view-wise robust Fréchet mean used in the mean-to-mean stabilizer (see Eq. (18) and Eq. (19)). We run a few iterations of tangent-space descent on the hypersphere (Bonnabel, 2013).

**Initialization.** Given samples $\{\mathbf{x}_\mathbf{u}\} \subset \mathbb{S}^{D-1}$ and weights $\{\bar{w}_\mathbf{u}\}$, we initialize with the normalized Euclidean mean:

$$\boldsymbol{\mu}^{(0)} = \text{normalize}\left(\sum_{\mathbf{u} \in \Omega} \bar{w}_\mathbf{u} \mathbf{x}_\mathbf{u}\right). \tag{45}$$

**Log-map on $\mathbb{S}^{D-1}$.** At iteration $t$, define $\cos\theta_\mathbf{u} = \langle\boldsymbol{\mu}^{(t)}, \mathbf{x}_\mathbf{u}\rangle$ and

$$\theta_\mathbf{u} = \arccos(\text{clamp}(\cos\theta_\mathbf{u}, -1 + \epsilon_c, 1 - \epsilon_c)). \tag{46}$$

The log-map at $\boldsymbol{\mu}^{(t)}$ is

$$\log_{\boldsymbol{\mu}^{(t)}}(\mathbf{x}_\mathbf{u}) = \frac{\theta_\mathbf{u}}{\sin\theta_\mathbf{u} + \epsilon}\left(\mathbf{x}_\mathbf{u} - \cos\theta_\mathbf{u}\boldsymbol{\mu}^{(t)}\right), \tag{47}$$

where $\epsilon_c$ clamps $\cos\theta_\mathbf{u}$ for numerical stability (we use $\epsilon_c = 10^{-6}$), and $\epsilon$ avoids division by zero (we use $\epsilon = 10^{-6}$).

**Huber reweighting.** The Huber penalty in Eq. (18) yields a robust coefficient

$$a_{\mathbf{u}} = \min\left(1, \frac{\delta}{\theta_{\mathbf{u}} + \epsilon}\right), \tag{48}$$

which down-weights samples with large geodesic errors.

**Tangent-space descent and retraction.** We compute the descent direction

$$\mathbf{g}^{(t)} = \sum_{\mathbf{u} \in \Omega} \bar{w}_{\mathbf{u}} \, a_{\mathbf{u}} \, \log_{\boldsymbol{\mu}^{(t)}}(\mathbf{x}_{\mathbf{u}}), \tag{49}$$

and update with a simple retraction:

$$\boldsymbol{\mu}^{(t+1)} = \text{normalize}\left(\boldsymbol{\mu}^{(t)} - \eta \, \mathbf{g}^{(t)}\right). \tag{50}$$

**Hyperparameters.** We use a small fixed number of iterations (e.g., 5). We set $\eta$ to a constant step size. We find this solver stable in practice.

**Differentiation.** We unroll the fixed number of iterations and backpropagate through the updates when computing $\mu_v^s$. We detach the VCD weights when forming $\bar{w}_{\mathbf{u}}$ to avoid degenerate solutions (i.e., $\bar{w}_{\mathbf{u}} \leftarrow \text{stopgrad}(\bar{w}_{\mathbf{u}})$). This mean computation is used only for the view-wise statistic and does not modify the CUDA rasterizer.

### B.4. LIC: Manifold-Aware Local Contrast on 3D Gaussians

LIC regularizes the *3D semantic field* on the Gaussian set. It enforces local consistency within regions and separation across nearby regions. Let $\mathcal{G} = \{g_i\}_{i=1}^N$. Each Gaussian has center $\mathbf{x}_i \in \mathbb{R}^3$ and a normalized feature $\mathbf{f}_i \in \mathbb{S}^{D-1}$.

**In-batch sampling.** At each LIC step, we sample $B$ Gaussians $\mathcal{I} = \{i_1, \ldots, i_B\}$. We stack their features and positions as $\mathbf{F} \in \mathbb{R}^{B \times D}$ and $\mathbf{X} \in \mathbb{R}^{B \times 3}$.

**Similarity on the hypersphere.** Features lie on $\mathbb{S}^{D-1}$, so we use cosine similarity with temperature $T$:

$$s_{ab} = \frac{\langle \mathbf{f}_{i_a}, \mathbf{f}_{i_b} \rangle}{T}. \tag{51}$$

**Neighborhood positives and negatives.** We define positives by a 3D radius $r$:

$$\mathcal{P}(i_a) = \{i_b \in \mathcal{I} \mid i_b \neq i_a, \, \|\mathbf{x}_{i_a} - \mathbf{x}_{i_b}\|_2 < r\}. \tag{52}$$

All remaining in-batch samples are treated as negatives:

$$\mathcal{N}(i_a) = \mathcal{I} \setminus (\mathcal{P}(i_a) \cup \{i_a\}). \tag{53}$$

**Hard negatives near boundaries.** Nearby negatives often lie across a semantic boundary. They are informative for sharpening transitions. We upweight nearby-but-nonpositive samples:

$$\mathcal{H}(i_a) = \{i_b \in \mathcal{N}(i_a) \mid \|\mathbf{x}_{i_a} - \mathbf{x}_{i_b}\|_2 < \eta r\}, \tag{54}$$

where $\eta > 1$ is a margin factor (we use $\eta = 2$). We add a constant bias to their logits:

$$\tilde{s}_{ab} = s_{ab} + \log(1 + \gamma) \cdot \mathbb{1}[i_b \in \mathcal{H}(i_a)], \tag{55}$$

where $\gamma \geq 0$ controls the strength.

**Multi-positive InfoNCE.** For each anchor $i_a$, we use a multi-positive InfoNCE objective:

$$\mathcal{L}_{\text{LIC}}(i_a) = -\log \frac{\sum\limits_{i_b \in \mathcal{P}(i_a)} \exp(\tilde{s}_{ab})}{\sum\limits_{i_c \in \mathcal{I} \setminus \{i_a\}} \exp(\tilde{s}_{ac})}. \tag{56}$$

Anchors with too few positives can be unstable. We keep only anchors with at least $m$ positives: $\mathcal{V} = \{i_a \in \mathcal{I} \mid |\mathcal{P}(i_a)| \geq m\}$ (we use $m{=}1$). We then average

$$\mathcal{L}_{\text{LIC}} = \frac{1}{|\mathcal{V}|} \sum_{i_a \in \mathcal{V}} \mathcal{L}_{\text{LIC}}(i_a). \tag{57}$$

In implementation, we compute Eq. (56) with $\log \sum \exp(\cdot)$ for numerical stability. We mask self-similarity by setting diagonal logits to $-\infty$.

**Why LIC is manifold-aware.** Cosine similarity is consistent with hyperspherical geometry. It depends only on angles between features. VFM also operates on $\mathbb{S}^{D-1}$. LIC therefore promotes *local angular smoothness* within surfaces. It also encourages *angular separation* across nearby regions.

**Complexity and scheduling.** Computing in-batch pairwise logits is $O(B^2)$. We therefore evaluate LIC intermittently, e.g., every $K_{\text{lic}}$ iterations. We use a moderate batch size $B$ (e.g., 1024–4096) to balance runtime and regularization strength.

### B.5. Additional Properties of VFM: Robustness and Influence

We provide a concise robustness interpretation of the Huberized Fréchet mean in VFM. Recall the objective: $\min_{\boldsymbol{\mu} \in \mathbb{S}^{D-1}} \sum_i \bar{w}_i \, \rho_\delta(\theta_i)$ with $\theta_i = d_{\mathbb{S}}(\boldsymbol{\mu}, \mathbf{x}_i)$.

**Proposition B.1 (Bounded influence via Huber reweighting).** Let $\boldsymbol{\mu}$ be a current estimate on $\mathbb{S}^{D-1}$ and define $\theta_i = d_{\mathbb{S}}(\boldsymbol{\mu}, \mathbf{x}_i)$. Consider the tangent-space descent direction in Eq. (49):

$$\mathbf{g}(\boldsymbol{\mu}) = \sum_{i=1}^{n} \bar{w}_i \, c_i \, \log_{\boldsymbol{\mu}}(\mathbf{x}_i), \qquad c_i = \min\left(1, \frac{\delta}{\theta_i + \epsilon}\right). \tag{58}$$

Then each sample's contribution to $\mathbf{g}(\boldsymbol{\mu})$ has bounded magnitude:

$$\left\| \bar{w}_i \, c_i \, \log_{\boldsymbol{\mu}}(\mathbf{x}_i) \right\|_2 \leq \bar{w}_i \cdot \delta, \tag{59}$$

up to numerical constants controlled by $\epsilon$.

*Sketch.* On $\mathbb{S}^{D-1}$, $\| \log_{\boldsymbol{\mu}}(\mathbf{x}_i) \|_2 = \theta_i$. Thus the per-sample magnitude is $\bar{w}_i \, c_i \, \theta_i \leq \bar{w}_i \min(\theta_i, \delta) \leq \bar{w}_i \delta$.

**Implication.** Eq. (59) shows that far-away samples ($\theta_i \gg \delta$), which often correspond to mixed-depth/occlusion outliers, cannot dominate the update direction: their influence is capped by $\delta$ rather than growing linearly with $\theta_i$ (as in $\ell_2$ geodesic aggregation). This is the main reason VFM remains stable even when a subset of pixels carry unreliable semantics.

**Remark (Why not plain Fréchet mean).** If we replace $\rho_\delta$ by $\frac{1}{2}\theta^2$, then $c_i \equiv 1$ and the contribution magnitude becomes $\bar{w}_i \theta_i$, which grows unbounded as $\theta_i$ increases. Hence, outliers can disproportionately steer the mean, yielding view-inconsistent semantics.

### B.6. Implementation-Notation Alignment

We clarify how the quantities in the paper correspond to the renderer outputs in our implementation.

**Opacity and moments.** In the 3DGS rasterizer, each pixel accumulates transmittance-weighted contributions along the ray. The returned maps correspond to:

- Accumulated opacity $A(\mathbf{u})$ in Eq. (36): `render_opacity` (or `opacity`) in `render_pkg`. Numerically, it equals the composited weight sum $\sum_k T_k \alpha_k = 1 - T_{\text{end}}$.

- First depth moment $\mathbf{D}^{(1)}(\mathbf{u})$ in Eq. (37): `render_depth` (or `depth`) in `render_pkg`. In our renderer, this map is accumulated as $\sum_k T_k \alpha_k z_k$ (not yet divided by $A$).

- Second depth moment $\mathbf{D}^{(2)}(\mathbf{u})$ in Eq. (37): `render_depth2` (or `depth2`) in `render_pkg`, accumulated as $\sum_k T_k \alpha_k z_k^2$.

Therefore, the expected depth and variance in Eq. (43) are computed by dividing the moments by $A(\mathbf{u}) + \epsilon$, matching our VCD implementation.

**Detaching visibility weights.** We set $\mathcal{W}(\mathbf{u})$ to be non-trainable during backpropagation by `detach()`. Without this, the model can reduce the loss by shrinking weights (e.g., lowering opacity) rather than improving semantics. This is a standard safeguard in reweighting-based distillation.

**Numerical stability for spherical distances.** For all spherical distances $d_{\mathbb{S}}(\mathbf{a}, \mathbf{b}) = \arccos(\langle \mathbf{a}, \mathbf{b} \rangle)$, we clamp cosine values to $[-1 + \epsilon, \, 1 - \epsilon]$ before calling `acos`. This prevents NaNs due to floating-point overshoot and is used consistently in both VFM and evaluation metrics.

**Where VFM is applied.** We apply VFM to $\ell_2$-normalized teacher and rendered embeddings. In implementation, we normalize feature maps channel-wise (per pixel) before computing cosine, angle, and geodesic loss. VCD provides per-pixel visibility weights, which are detached and broadcast (or resized if needed) to the feature resolution. These weights are used to compute a weighted average of the pixel-wise loss, and to weight samples in the view-wise Fréchet-mean stabilizer.

### B.7. Derivation of the Euclidean–hyperspherical gradient mismatch.

Let the standard rendered semantic feature before normalization be

$$F = \sum_k w_k h_k, \tag{60}$$

where $h_k$ is the semantic feature of the $k$-th Gaussian and $w_k$ is its alpha-compositing weight. For the semantic feature gradient, $w_k$ is independent of $h_k$. We define the normalized rendered feature and the normalized teacher feature as

$$x = \hat{z} = \frac{F}{\|F\|}, \qquad y = z^T, \qquad \|x\| = \|y\| = 1. \tag{61}$$

Let

$$c = \langle x, y \rangle = \cos\theta, \qquad \theta = d_{\mathbb{S}}(x, y) = \arccos(c). \tag{62}$$

We first compute the Jacobian of the normalization operation. Let $r = \|F\|$, so $x = F/r$. The differential of $x$ with respect to $F$ is

$$dx = d\left(\frac{F}{\|F\|}\right) = \frac{1}{r}\left(I - xx^\top\right) dF. \tag{63}$$

Define the tangent-space projection matrix at $x$ as

$$P_x = I - xx^\top. \tag{64}$$

Then

$$dx = \frac{1}{\|F\|} P_x dF. \tag{65}$$

Since

$$F = \sum_k w_k h_k, \qquad dF = w_k dh_k, \tag{66}$$

for any loss $L(x)$, the chain rule gives

$$\nabla_{h_k} L = \frac{w_k}{\|F\|} P_x \nabla_x L. \tag{67}$$

Therefore, the difference between Euclidean and hyperspherical losses comes from their projected gradients on the tangent space of the unit sphere.

**Euclidean loss.**   The normalized Euclidean loss is

$$L_E = \frac{1}{2}\|x - y\|^2. \tag{68}$$

Because $\|x\| = \|y\| = 1$, it can also be written as

$$L_E = \frac{1}{2}\left(\|x\|^2 + \|y\|^2 - 2\langle x, y\rangle\right) = 1 - \langle x, y\rangle = 1 - c. \tag{69}$$

The ambient Euclidean gradient with respect to $x$ is

$$\nabla_x L_E = x - y. \tag{70}$$

After passing through the normalization layer, only the tangent component remains:

$$P_x \nabla_x L_E = P_x(x - y) = P_x x - P_x y. \tag{71}$$

Since

$$P_x x = (I - xx^\top)x = x - x(x^\top x) = x - x = 0, \tag{72}$$

we have

$$P_x \nabla_x L_E = -P_x y. \tag{73}$$

Moreover,

$$P_x y = (I - xx^\top)y = y - \langle x, y\rangle x = y - cx. \tag{74}$$

Thus,

$$\nabla_{h_k} L_E = -\frac{w_k}{\|F\|}\left(y - cx\right). \tag{75}$$

The vector $y - cx$ lies in the tangent space at $x$, because

$$x^\top(y - cx) = x^\top y - cx^\top x = c - c = 0. \tag{76}$$

Its norm is

$$\|y - cx\|^2 = \|y\|^2 - 2c\langle x, y\rangle + c^2\|x\|^2 = 1 - 2c^2 + c^2 = 1 - c^2 = \sin^2\theta. \tag{77}$$

Therefore,

$$\|\nabla_{h_k} L_E\| = \frac{|w_k|}{\|F\|}\sin\theta. \tag{78}$$

**Geodesic loss on the hypersphere.**   The squared geodesic loss is

$$L_S = \frac{1}{2}d_{\mathbb{S}}(x, y)^2 = \frac{1}{2}\theta^2, \qquad \theta = \arccos(\langle x, y\rangle). \tag{79}$$

Since

$$c = \langle x, y\rangle, \tag{80}$$

we have

$$\frac{\partial\theta}{\partial c} = -\frac{1}{\sqrt{1 - c^2}} = -\frac{1}{\sin\theta}. \tag{81}$$

Also,

$$\nabla_x c = \nabla_x\langle x, y\rangle = y. \tag{82}$$

Thus,

$$\nabla_x \theta = \frac{\partial\theta}{\partial c}\nabla_x c = -\frac{1}{\sin\theta}y. \tag{83}$$

The ambient gradient of $L_S$ is therefore

$$\nabla_x L_S = \theta\nabla_x\theta = -\frac{\theta}{\sin\theta}y. \tag{84}$$

After the normalization layer, we project this gradient onto the tangent space:

$$P_x \nabla_x L_S = -\frac{\theta}{\sin\theta} P_x y = -\frac{\theta}{\sin\theta} (y - cx). \tag{85}$$

Using the chain rule with respect to $h_k$, we obtain

$$\nabla_{h_k} L_S = -\frac{w_k}{\|F\|} \frac{\theta}{\sin\theta} (y - cx). \tag{86}$$

Therefore, the Euclidean and geodesic gradients have the same tangent-space direction:

$$-(y - cx), \tag{87}$$

but they differ by a scalar factor:

$$\nabla_{h_k} L_S = \frac{\theta}{\sin\theta} \nabla_{h_k} L_E. \tag{88}$$

Taking norms gives

$$\frac{\|\nabla_{h_k} L_S\|}{\|\nabla_{h_k} L_E\|} = \frac{\theta}{\sin\theta}. \tag{89}$$

**When does the mismatch matter?** For $\theta \in (0, \pi)$, we have

$$\frac{\theta}{\sin\theta} > 1. \tag{90}$$

Moreover, this factor is monotonically increasing. Indeed,

$$\frac{d}{d\theta}\left(\frac{\theta}{\sin\theta}\right) = \frac{\sin\theta - \theta\cos\theta}{\sin^2\theta}. \tag{91}$$

For $\theta \in (0, \pi/2]$, $\sin\theta - \theta\cos\theta > 0$. For $\theta \in (\pi/2, \pi)$, $\cos\theta < 0$, so $\sin\theta - \theta\cos\theta > 0$ also holds. Therefore,

$$\frac{\theta}{\sin\theta} \tag{92}$$

increases with angular error $\theta$.

This result shows that the mismatch is negligible when the angular error is small, because

$$\frac{\theta}{\sin\theta} = 1 + \frac{\theta^2}{6} + O(\theta^4), \qquad \theta \to 0. \tag{93}$$

However, when $\theta$ becomes large, the Euclidean loss provides a much weaker corrective signal. Specifically, the gradient magnitude of the Euclidean loss is proportional to $\sin\theta$, while that of the squared geodesic loss is proportional to $\theta$:

$$\|\nabla_{h_k} L_E\| = \frac{|w_k|}{\|F\|} \sin\theta, \qquad \|\nabla_{h_k} L_S\| = \frac{|w_k|}{\|F\|} \theta. \tag{94}$$

Thus, for high-error pixels, such as boundary, occlusion, or mixed-depth pixels, the geodesic loss supplies stronger corrective gradients. This is exactly the regime where Euclidean–hyperspherical mismatch materially affects optimization and prediction quality.

**Extension to the Huberized geodesic loss.** In our VFM objective, the geodesic discrepancy can be Huberized:

$$L_\rho = \rho_\delta(\theta), \tag{95}$$

where

$$\rho_\delta(\theta) = \begin{cases} \frac{1}{2}\theta^2, & \theta \leq \delta, \\ \delta(\theta - \frac{1}{2}\delta), & \theta > \delta. \end{cases} \tag{96}$$

Its derivative is

$$\rho'_\delta(\theta) = \begin{cases} \theta, & \theta \le \delta, \\ \delta, & \theta > \delta. \end{cases} \tag{97}$$

Following the same derivation, we obtain

$$\nabla_{h_k} L_\rho = -\frac{w_k}{\|F\|} \frac{\rho'_\delta(\theta)}{\sin\theta} (y - cx). \tag{98}$$

Thus, in the quadratic region $\theta \le \delta$, the Huberized loss behaves exactly like the squared geodesic loss:

$$\nabla_{h_k} L_\rho = \nabla_{h_k} L_S. \tag{99}$$

In the linear region $\theta > \delta$, its gradient magnitude becomes bounded:

$$\|\nabla_{h_k} L_\rho\| = \frac{|w_k|}{\|F\|}\delta. \tag{100}$$

Therefore, the Huberized geodesic loss preserves the manifold-aware tangent direction while preventing extreme outliers from dominating the optimization. This explains why VFM is both geometry-consistent and robust.

## C. Additional Experimental Results

This section presents additional qualitative results omitted from the main paper due to space limits. We include: (i) multi-view open-vocabulary segmentation on LERF, (ii) multi-view semantic segmentation on 3D-OVS, and (iii) scene editing enabled by the learned 3D semantic field. Unless noted otherwise, all visualizations are rendered from the optimized 3D Gaussians. We follow the same evaluation protocol as in the main paper.

### C.1. Multi-view Segmentation Results on the LERF Dataset

Fig. 5 shows additional multi-view open-vocabulary segmentation results on LERF. Each scene is rendered from multiple viewpoints. We also evaluate multiple text queries per scene (see the legend). Rh-3DGS keeps the masks stable across views. It also preserves small objects and thin structures. Examples include small tabletop items and fine boundaries between neighboring objects.

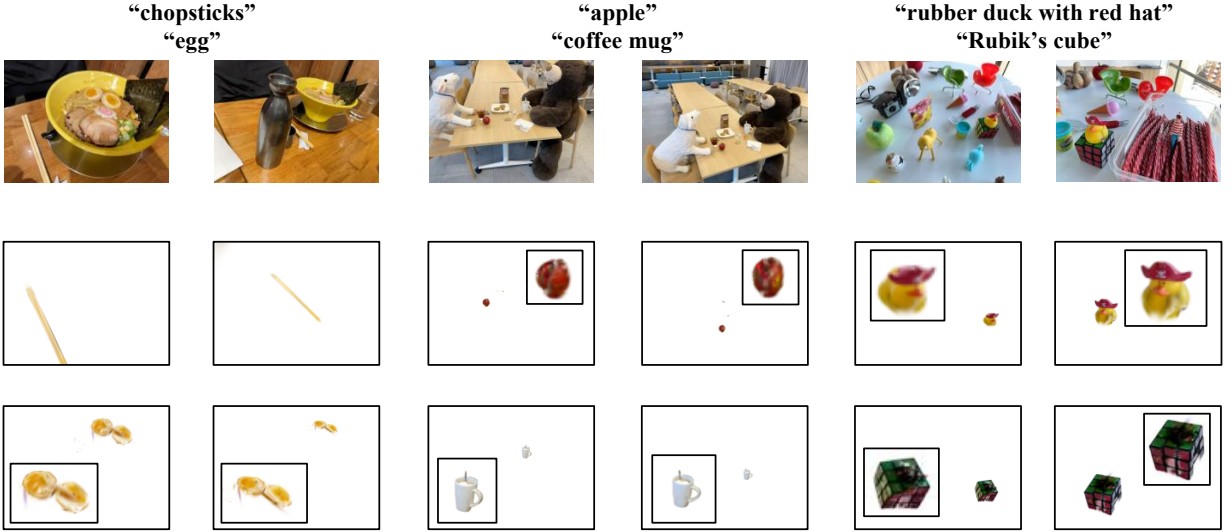

*Figure 5.* **LERF multi-view open-vocabulary segmentation.** Each row is one scene and a set of text queries (legend). Columns show different viewpoints of the same scene. Rh-3DGS produces view-consistent masks and preserves small objects and thin boundaries.

## C.2. Multi-view Semantic Segmentation Results on the 3D-OVS Dataset

Fig. 6 presents additional multi-view semantic segmentation results on 3D-OVS. We visualize multiple viewpoints for the same scene. Rh-3DGS produces consistent predictions under occlusions and depth discontinuities. This reduces flickering masks across views.

## C.3. Scene Editing Results

Fig. 7 shows text-guided scene editing enabled by our 3D semantic field. We first localize the target region using a text query. We then apply a localized edit and render the edited scene from multiple viewpoints. The edits remain consistent across views. We show three examples: deleting a green apple, recoloring a red toy chair to yellow, and resizing a red apple.

# D. Extended Ablations

This section reports additional ablations omitted from the main paper due to space limits. We include: (i) VCD weight component analysis, (ii) loss-weight sensitivity, (iii) LIC positive-pair design, and (iv) key hyper-parameter sweeps. Unless stated otherwise, we use the default setting in the main paper.

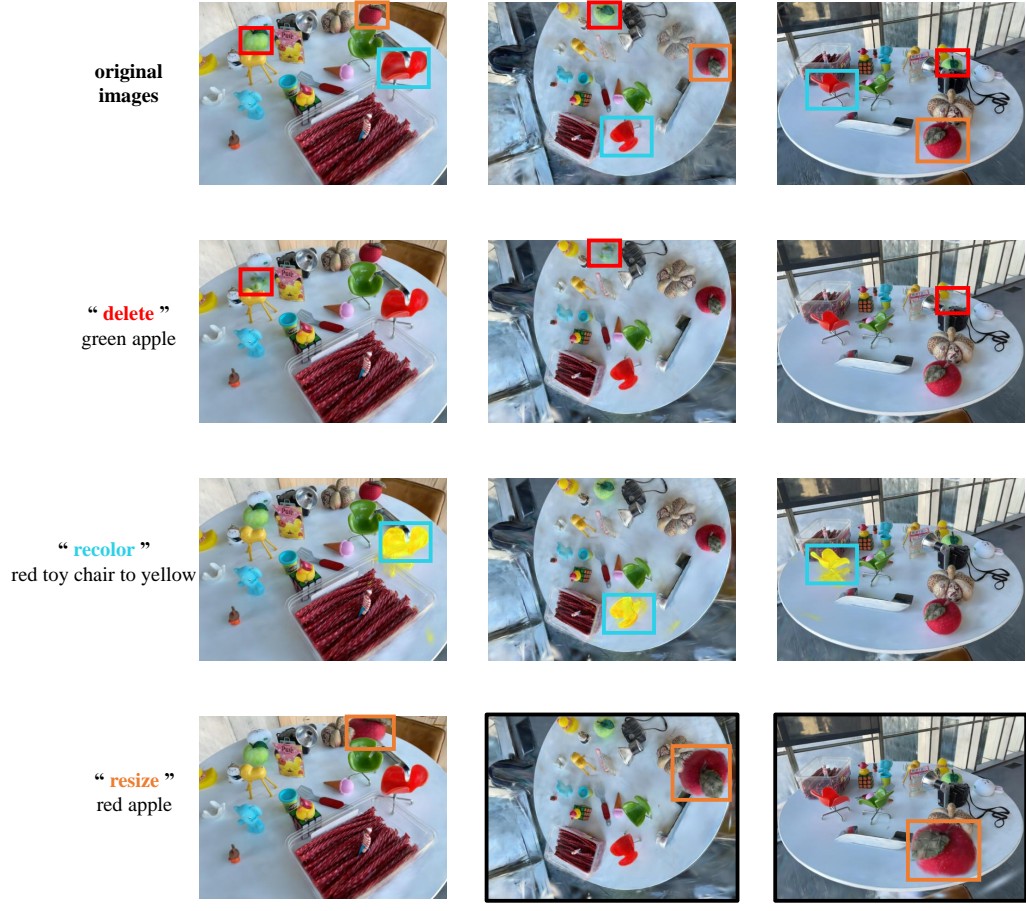

*Figure 7.* **Scene editing with the learned 3D semantic field.** We show the original renderings, the localized semantic region, and the edited renderings from multiple viewpoints. Edits are view-consistent, including *delete*, *recolor*, and *resize* operations.

## D.1. Ablation of VCD Weight Components

VCD builds a reliability map $\mathcal{W} = W_{op} \cdot W_{edge} \cdot W_{var}$. It combines opacity gating $W_{op}$, depth-edge suppression $W_{edge}$, and ray-variance calibration $W_{var}$. We remove one factor at a time and keep the others unchanged. We always use $\mathcal{W} \leftarrow \text{stopgrad}(\mathcal{W})$.

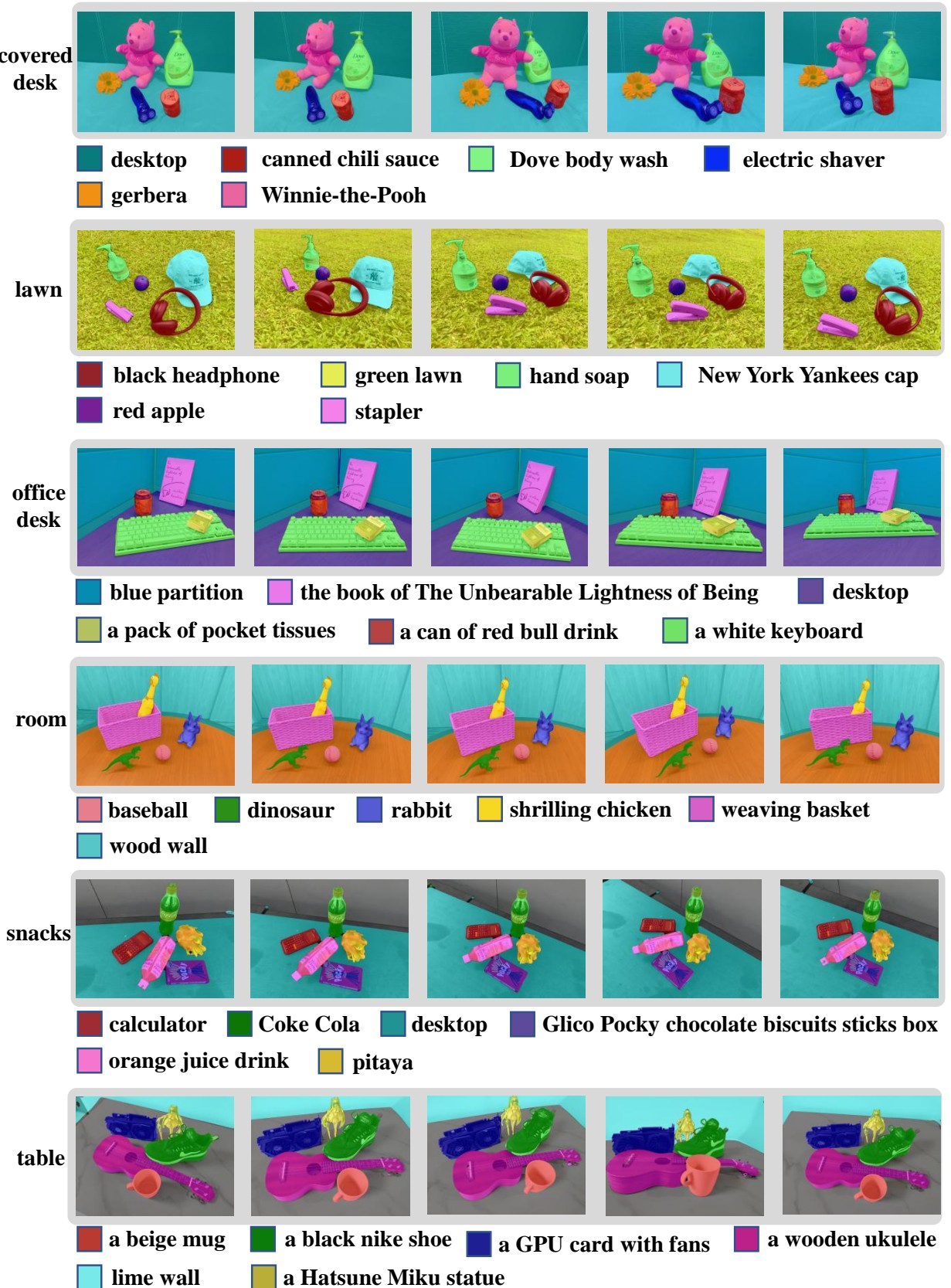

*Figure 6.* **3D-OVS multi-view semantic segmentation.** Rows show different viewpoints of the same scene. Columns show RGB, our prediction, and the reference (and an optional baseline, if included). Rh-3DGS remains consistent under occlusions and depth changes.

*Table 10.* **Ablation of VCD weight components** on LERF (figurines).

| $W_{op}$ | $W_{edge}$ | $W_{var}$ | mIoU ↑ |
|:---:|:---:|:---:|:---:|
| ✓ | ✓ | ✓ | 81.62 |
| ✗ | ✓ | ✓ | 76.89 |
| ✓ | ✗ | ✓ | 77.02 |
| ✓ | ✓ | ✗ | 78.01 |

**Discussion.** All three factors contribute to performance. Removing any single factor causes a clear drop. $W_{edge}$ and $W_{var}$ target occlusion boundaries and mixed-depth rays. They are important for suppressing boundary noise. $W_{op}$ stabilizes low-coverage pixels and empty regions. It prevents unreliable supervision from sparse rays.

### D.2. Sensitivity to Loss Weights

We vary $\lambda_{\text{VFM}}$ and $\lambda_{\text{LIC}}$ while keeping other settings fixed. Fig. 8 shows the curves. The performance is stable around our default configuration. This suggests that Rh-3DGS is not sensitive to small changes in these weights.

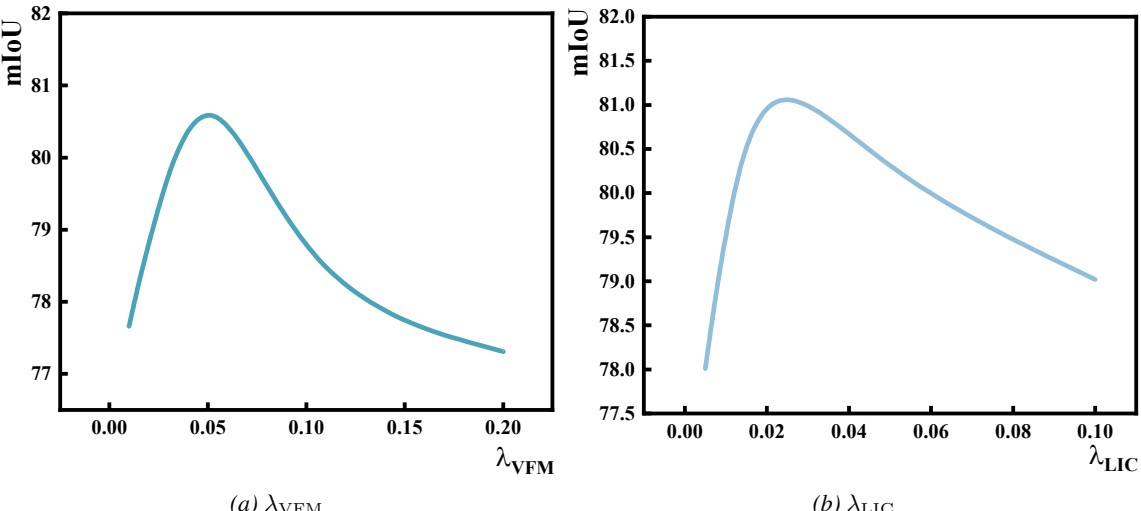

*(a) $\lambda_{\text{VFM}}$*  *(b) $\lambda_{\text{LIC}}$*

*Figure 8.* **Sensitivity to loss weights on LERF (figurines).** We sweep $\lambda_{\text{VFM}}$ and $\lambda_{\text{LIC}}$ and report mIoU. The best region is around our default setting.

### D.3. Ablation on LIC Positive-pair Construction

LIC depends on how we build positive pairs. We compare two strategies. (i) **Instance clustering:** we cluster Gaussians into $K$ groups (e.g., K-means on normalized features). We sample positives within the same cluster. (ii) **Spatial radius only:** we define positives by a 3D radius $r$, without instance identity. We also vary $K$ and $r$.

*Table 11.* **LIC positives: instance clustering vs. spatial radius** on LERF (figurines).

| Positive definition | Setting | mIoU ↑ |
|:---|:---:|:---:|
| Instance clustering | $K = 20$ | 81.62 |
| Instance clustering | $K = 50$ | 80.20 |
| Spatial radius only | $r = 0.4$ | 78.01 |
| Spatial radius only | $r = 0.6$ | 76.89 |

**Discussion.** Instance clustering works better than spatial radius only. Pure proximity can connect nearby but different

objects. This introduces false-positive constraints. A moderate number of clusters ($K = 20$) performs best in our setting. A larger $K$ may fragment instances and reduce positive coverage.

### D.4. Key Hyper-parameter Sweeps

We sweep two key hyper-parameters. We vary the opacity threshold $\tau_{op}$ in $W_{op}$. We also vary the Huber parameter $\delta$ in VFM. Fig. 9 reports mIoU under these changes. Rh-3DGS remains stable in a reasonable range.

**Discussion.** A large $\tau_{op}$ removes too many pixels and weakens supervision. A small $\tau_{op}$ filters less and allows more noise. For VFM, a moderate $\delta$ provides a good robustness–discriminability trade-off. A very small $\delta$ can over-suppress hard but valid samples. A very large $\delta$ reduces robustness to outliers.

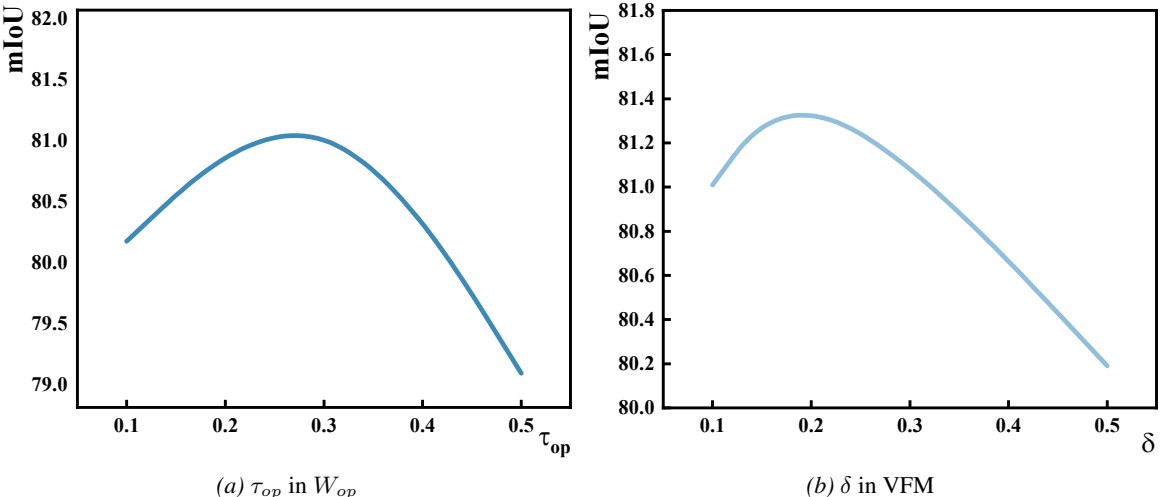

*(a) $\tau_{op}$ in $W_{op}$*          *(b) $\delta$ in VFM*

*Figure 9.* **Key hyper-parameter sweeps on LERF (figurines).** We sweep $\tau_{op}$ and $\delta$ and report mIoU. The method remains stable around our default setting.

## E. Pseudocode

---

**Algorithm 1** VFM Tangent Descent on $\mathbb{S}^{D-1}$

---

**Require:** Spherical samples $\{x_u\}_{u=1}^{M} \subset \mathbb{S}^{D-1}$, weights $\{w_u\}$, Huber threshold $\delta$, steps $K$, step size $\eta_{\text{vfm}}$, $\epsilon$.
**Ensure:** Robust mean $\mu \in \mathbb{S}^{D-1}$.
1: $\bar{w}_u \leftarrow w_u/(\sum_j w_j + \epsilon)$.
2: $\mu \leftarrow \text{NORMALIZE}\left(\sum_u \bar{w}_u x_u\right)$.
3: **for** $k = 1$ **to** $K$ **do**
4:      $\hat{c}_u \leftarrow \text{CLAMP}(\langle x_u, \mu \rangle, -1 + \epsilon, 1 - \epsilon)$.
5:      $\theta_u \leftarrow \arccos(\hat{c}_u)$;    $s_u \leftarrow \sqrt{1 - \hat{c}_u^2} + \epsilon$.
6:      $\log_\mu(x_u) \leftarrow (\theta_u/(s_u + \epsilon)) \cdot (x_u - \hat{c}_u \mu)$.
7:      $a_u \leftarrow \min\left(1, \delta/(\theta_u + \epsilon)\right)$ {Huber reweight}
8:      $g \leftarrow \sum_u \bar{w}_u a_u \log_\mu(x_u)$.
9:      $\mu \leftarrow \text{NORMALIZE}(\mu - \eta_{\text{vfm}} g)$ {retraction}
10: **end for**
11: **return** $\mu$.

---

---

**Algorithm 2** Visibility-Calibrated Distillation (VCD)

---

**Require:** Accumulated opacity $A \in \mathbb{R}^{H \times W}$, depth moments $D^{(1)}, D^{(2)} \in \mathbb{R}^{H \times W}$ (optional), $\Theta_{\text{vcd}} = \{\tau_{op}, s, \gamma, \beta, q\}$, quantile normalization $\mathcal{N}_q(\cdot)$, $\epsilon$.
**Ensure:** Reliability weights $\mathcal{W} \in \mathbb{R}^{H \times W}$.
  1: $W_{op} \leftarrow \sigma\big(s\,(A - \tau_{op})\big)$ {opacity gate}
  2: **if** $D^{(1)}$ is not available **then**
  3:  $\mathcal{W} \leftarrow W_{op}$
  4:  **return** STOPGRAD($\mathcal{W}$)
  5: **end if**
  6: $\bar{D} \leftarrow D^{(1)}/(A + \epsilon)$ {expected depth}
  7: $G \leftarrow \|\nabla \log(\bar{D} + \epsilon)\|$ {Sobel on log-depth}
  8: $W_{edge} \leftarrow \exp\big(-\gamma \cdot \mathcal{N}_q(G)\big)$
  9: $\mathcal{W} \leftarrow W_{op} \odot W_{edge}$
 10: **if** $D^{(2)}$ is available **then**
 11:  $\mathbb{E}[z^2] \leftarrow D^{(2)}/(A + \epsilon)$
 12:  $\text{Var}(z) \leftarrow \big[\mathbb{E}[z^2] - \bar{D} \odot \bar{D}\big]_+$
 13:  $W_{var} \leftarrow \exp\big(-\beta \cdot \mathcal{N}_q(\text{Var}(z))\big)$
 14:  $\mathcal{W} \leftarrow \mathcal{W} \odot W_{var}$
 15: **end if**
 16: **return** STOPGRAD($\mathcal{W}$)

---

**Algorithm 3** LIC: Lightweight Consistency Contrast (one step)

---

**Require:** Centers $\{\mathbf{x}_i\}_{i=1}^N$, decoded features $\{\mathbf{h}_i\}_{i=1}^N$, radius $r$, hard factor $\kappa > 1$, hard weight $\gamma \geq 1$, temperature $T_c$, batch size $B$, patch size $P$, min positives $m$, (optional) pseudo-instance ids $\pi(i)$; if disabled, set $\pi(i) \equiv 1$.
**Ensure:** $\mathcal{L}_{\text{LIC}}$.
  1: **Patch-based sampling:** sample $B/P$ seed indices $\mathcal{S}$ uniformly from $\{1, \ldots, N\}$.
  2: For each seed $s \in \mathcal{S}$, sample $P$ Gaussians from its spatial neighborhood; collect $\mathcal{I} = \{i_1, \ldots, i_B\}$.
  3: $X \leftarrow [\mathbf{x}_{i_1}, \ldots, \mathbf{x}_{i_B}]$.
  4: $F \leftarrow [\tilde{\mathbf{h}}_{i_1}, \ldots, \tilde{\mathbf{h}}_{i_B}]$, where $\tilde{\mathbf{h}}_i = \mathbf{h}_i/\|\mathbf{h}_i\|_2$.
  5: Compute pairwise distances $\Delta_{ab} \leftarrow \|\mathbf{x}_{i_a} - \mathbf{x}_{i_b}\|_2$.
  6: Compute logits $S_{ab} \leftarrow \langle \tilde{\mathbf{h}}_{i_a}, \tilde{\mathbf{h}}_{i_b} \rangle / T_c$.
  7: Mask diagonal: $S_{aa} \leftarrow -\infty$.
  8: Positive mask $P_{ab} \leftarrow [\Delta_{ab} < r] \wedge [\pi(i_a) = \pi(i_b)] \wedge [a \neq b]$.
  9: Hard-neg mask $H_{ab} \leftarrow [\Delta_{ab} < \kappa r] \wedge \neg P_{ab} \wedge [a \neq b]$.
 10: $S_{ab} \leftarrow S_{ab} + \log(\gamma) \cdot H_{ab}$.
 11: Valid anchors $V \leftarrow \{a \mid \sum_b P_{ab} \geq m\}$.
 12: **if** $|V| = 0$ **then**
 13:  **return** $0$.
 14: **end if**
 15: **for** each $a \in V$ **do**
 16:  $\log \text{num}_a \leftarrow \log \sum_{b:P_{ab}=1} \exp(S_{ab})$.
 17:  $\log \text{den}_a \leftarrow \log \sum_{c \neq a} \exp(S_{ac})$.
 18:  $\ell_a \leftarrow -(\log \text{num}_a - \log \text{den}_a)$.
 19: **end for**
 20: **return** $\mathcal{L}_{\text{LIC}} \leftarrow \frac{1}{|V|} \sum_{a \in V} \ell_a$.

---

---

**Algorithm 4** Rh-3DGS Training Loop

---

**Require:** Posed images $\{I_v\}$, teacher feature extractor $\Phi(\cdot)$, initial Gaussians $\mathcal{G} = \{g_i\}_{i=1}^{N}$, iterations $T$, VCD params $\Theta_{\text{vcd}}$, VFM steps $K$ and step size $\eta_{\text{vfm}}$, LIC params $(r, \kappa, \gamma, T_c, B, m)$, loss weights $(\lambda_{\text{rgb}}, \lambda_{\text{sem}}, \lambda_{\text{con}})$, LIC schedule $(t_0, \Delta)$.

**Ensure:** Optimized Gaussians $\mathcal{G}$ (and optional decoder $\psi$).

1: **for** $t = 1$ **to** $T$ **do**
2:      Sample a training view $v$ and load $I_v$.
3:      Query teacher feature map $Z_v \leftarrow \Phi(I_v)$.
4:      Align resolution if needed: $Z_v \leftarrow \text{RESIZE}(Z_v)$.
5:      Render from Gaussians: $(\hat{I}_v, \hat{Z}_v, A_v, D_v^{(1)}, D_v^{(2)}) \leftarrow \text{RENDER}(\mathcal{G}, v)$.
6:      $\mathcal{L}_{\text{rgb}} \leftarrow \text{RGBLOSS}(\hat{I}_v, I_v)$.
7:      $W_v \leftarrow \text{VCD}(A_v, D_v^{(1)}, D_v^{(2)}; \Theta_{\text{vcd}})$ {stop-grad on $W_v$}
8:      $\mathcal{L}_{\text{sem}} \leftarrow \text{VFM-LOSS}(\hat{Z}_v, Z_v, W_v; K, \eta_{\text{vfm}})$.
9:      **if** $t \geq t_0$ **and** $t \bmod \Delta = 0$ **then**
10:         $\mathcal{L}_{\text{con}} \leftarrow \text{LIC}(\{\mathbf{x}_i, \mathbf{f}_i\}_{i=1}^{N}; r, \kappa, \gamma, T_c, B, m)$.
11:      **else**
12:         $\mathcal{L}_{\text{con}} \leftarrow 0$.
13:      **end if**
14:      $\mathcal{L} \leftarrow \lambda_{\text{rgb}}\mathcal{L}_{\text{rgb}} + \lambda_{\text{sem}}\mathcal{L}_{\text{sem}} + \lambda_{\text{con}}\mathcal{L}_{\text{con}}$.
15:      Update $(\mathcal{G}, \psi)$ by one optimizer step using $\nabla \mathcal{L}$.
16: **end for**
17: **return** $\mathcal{G}$.

---

