# OpenReview forum: "Rh-3DGS: Robust Open-Vocabulary Scene Understanding via Riemannian Huber Distillation and Manifold-Aware Sampling"
_ICML.cc/2026/Conference — ICML 2026 regular_

### Official Review · Reviewer_KuQ8 · 2026-03-06

**Soundness:** 3
**Presentation:** 3
**Significance:** 3
**Originality:** 3
**Overall Recommendation:** 4
**Confidence:** 4

**Summary:**

Rh-3DGS addresses a weakness in lifting 2D vision-language features (e.g., CLIP) into 3D Gaussian Splatting: Euclidean averaging of normalized embeddings ignores their hyperspherical geometry, leading to norm shrinkage and feature collapse. Treating all pixels equally during distillation further amplifies noise from occlusions and mixed-depth rays. The method introduces three components: VCD down-weights unreliable pixels using rasterization statistics (opacity, depth edges, ray variance). VFM replaces the Euclidean loss with a Huberized geodesic objective on the hypersphere, stabilized by aligning view-wise Fréchet means between teacher and rendered features. LIC enforces local semantic coherence via multi-positive InfoNCE over spatial neighborhoods of Gaussians, sharpening boundaries. On LERF, 3D-OVS, and ScanNet, the method outperforms baselines like LangSplat and OpenGaussian, with notable gains on boundary metrics (mBIoU), while running in real time.

**Compliance With Llm Reviewing Policy:**

Affirmed.

**Ethical Review Concerns:**

-

**Final Justification:**

The authors have addressed my concerns in their rebuttal.

**Key Questions For Authors:**

How much norm shrinkage actually happens in practice?
How much do angular relationships distort?
Do all baselines use exactly the same features? If not, what are the results with the same features?

**Limitations:**

To some degree. The preprocessing overhead from the SAM+CLIP extraction is not discussed.

**Strengths And Weaknesses:**

Strengths:
- The core idea is well-motivated. Euclidean averaging of normalized embeddings causes norm shrinkage, and this is clearly shown both mathematically and visually. Using geodesic losses on the hypersphere is principled fix grounded in Riemannian statistics.
- The results are strong and consistent across three benchmarks and multiple metrics. The boundary improvements stand out - mBIoU gains of over 12 points on LERF suggest the method genuinely addresses boundary bleeding rather than just improving bulk region accuracy. This is also shown in the qualitative examples.
- The appendix is thorough. It includes full hyperparameters, pseudocode for all three algorithms, and a clear mapping between notation and implementation. This makes reproduction substantially easier.

Weaknesses:
- The paper argues that the Euclidean-to-hyperspherical gap is important but never measures it directly. How much norm shrinkage actually happens in practice? How much do angular relationships distort? Simple diagnostics, such as feature norm histograms or angular deviation plots before and after fusion, would make the motivation much more convincing.
- Some baselines perform dramatically worse than others, which raises questions about fairness. It is unclear whether all methods use the same features (SAM+CLIP), resolution, and evaluation pipeline. Different methods often use different settings for SAM. The paper should state this explicitly; and run all baselines and the proposed method on exactly the same features per image.
- The pipeline figure (Fig. 2) is hard to understand without having read the full paper. It is full of abbreviations and symbols that are only defined later. A pipeline figure should give a rough understanding of the method before diving into the details. I would suggest redesigning it with more descriptive labels so it can stand on its own.
- The method is fast at rendering time, but it inherits the heavy preprocessing cost of extracting dense SAM and CLIP features from all training views. In practice, this can take hours. This is not unique to this paper - it is common in distillation-based methods - but it should be clearly acknowledged. Reporting total wall-clock time including preprocessing would give a more honest picture.

Overall: I like that paper but there are certain things that has to be clarified. Most importantly, the question whether all methods use exactly the same features should be discussed. If not, they should be rerun with the same features.

---

> ### Author Rebuttal · Authors · 2026-03-30
>
> We thank the reviewer for the positive evaluation of the paper's core motivation, empirical results, and reproducibility. We especially appreciate the reviewer's focus on two issues that indeed deserve to be made more explicit: (i) the direct measurement of the Euclidean–hyperspherical gap in practice, and (ii) the fairness of comparisons regarding teacher features/evaluation protocols. These are exactly the right areas to strengthen the current draft.
>
> **1.Directly measuring norm shrinkage and angular distortion**
>
> To address this concern, we added a direct diagnostic of the Euclidean–hyperspherical gap on the baseline model. We measure: the pre-normalization Euclidean fused norm (norm shrinkage), and the angular deviation between the Euclidean normalized mean and the weighted Fréchet mean.
> |Region|Avg. pre-norm norm of s|Avg. angular deviation|
> |---|---:|---:|
> |All pixels|0.957|16.16|
> |Boundary|0.924|18.12|
> |Non-boundary|0.965|15.66|
> |High depth-var|0.945|17.66|
> |Low depth-var|0.960| 15.77|
>
> The resulting statistics show that the gap is non-trivial in practice:
>
> **All pixels:** mean angular deviation=16.16°, mean pre-normalization norm=0.957;
>
> **Boundary vs. non-boundary:** 18.12° vs. 15.66°, and 0.924 vs. 0.965;
>
> **High-var vs. low-var:** 17.66° vs. 15.77°, and 0.945 vs. 0.960.
>
> These results directly support our motivation: Euclidean fusion does induce measurable norm shrinkage and directional distortion, and both effects are amplified on the difficult regions (boundaries/high depth variance) where Rh-3DGS shows the strongest boundary gains.
>
> **2.Do all baselines use the same features?**
>
> Yes. To clarify evaluation fairness: all compared methods use the same frozen teacher features extracted from SAM ViT-H and OpenCLIP ViT-B/16, with the same image resolution and the same evaluation pipeline. This is also consistent with our implementation details, where teacher supervision uses frozen SAM (ViT-H) and OpenCLIP (ViT-B/16), and teacher parameters remain fixed during training.
>
> **3.On the pipeline figure (Fig. 2)**
>
> Thank you for this suggestion. We have redesigned Fig. 2 as a clearer overview figure with larger descriptive labels and a simpler problem-to-method-to-output structure, so that it can be understood before reading the detailed method section. The redesigned figure is provided at the anonymous link: https://doi.org/10.6084/m9.figshare.31883308.
>
> **4.Preprocessing overhead**
>
> Thank you for this suggestion. We have added the end-to-end wall-clock time on the LERF figurines scene, including both dense SAM+CLIP preprocessing and training:
>
> |SAM+CLIP preprocessing|Training|Total|
> |---|:---:|:---:|
> |345min10s|29min20s|374min30s|
>
> This shows that, although Rh-3DGS runs in real time at inference, the overall pipeline still inherits the heavy teacher-feature extraction cost common to distillation-based methods.
>
> We thank the reviewer again for pointing out these important clarity issues.

---

> > ### Author Rebuttal · Reviewer_KuQ8 · 2026-04-01
> >
> > I am happy with the authors' response. I will improve my rating.

---

> > > ### Author Response · Authors · 2026-04-02
> > >
> > > Thank you very much for carefully reading our rebuttal and for your positive feedback. We sincerely appreciate that our response addressed your concerns, and we are grateful for your updated rating.

---

### Official Review · Reviewer_vqLc · 2026-03-12

**Soundness:** 3
**Presentation:** 3
**Significance:** 2
**Originality:** 2
**Overall Recommendation:** 4
**Confidence:** 4

**Summary:**

The paper proposes Rh-3DGS, a framework for open-vocabulary 3D scene understanding. The paper aims to address a problem: that existing methods ignore the hyperspherical geometry of normalized CLIP/SAM embeddings when lifting 2D features into 3D Gaussians. A central context assessed by the paper is the compounding problem of Euclidean feature collapse and unreliable per-pixel supervision near occlusions and mixed-depth boundaries. Rh-3DGS introduces three modules: Visibility-Calibrated Distillation (VCD), which computes per-pixel reliability weights from rasterization; Visibility-Weighted Frechet Mean (VFM), which performs Huberized geodesic distillation on the unit hypersphere; and Lightweight Consistency Contrast (LIC), a neighborhood-based contrastive regularizer applied directly to Gaussian semantics. Experiments on LERF, 3D-OVS, and ScanNet demonstrate consistent improvements, particularly on mBIoU.

**Compliance With Llm Reviewing Policy:**

Affirmed.

**Key Questions For Authors:**

- The paper reports VFM at 73.76 mIoU in Table 5, which is lower than both LIC alone (75.70) and the baseline + VCD alone (65.37). Within the VFM configuration, could the authors report the performance of 1) pixel-wise Huberized geodesic loss $L_{pix}$ alone (without the mean-to-mean stabilizer $L_mean$) and 2) the mean-to-mean stabilizer alone (with standard cosine loss for $L_{pix}$)? This decomposition is important because the Frechet mean stabilizer adds non-trivial computational cost and the paper's core theoretical claim about hyperspherical geometry is primarily realized in $L_mean$.

- Regarding the behavior of VCD weights during training, VCD weights $W_v(u)$ are computed from rasterization (opacity, depth gradient etc) and stopped from backpropagating gradients. Since these parameters change substantially as Gaussians are optimized, e.g., during the densification phase (iterations 500–15,000) when Gaussian count and positions shift rapidly, could the authors analyze how the distribution of VCD weights evolves during training? Specifically, what fraction of pixels are effectively masked (W < 0.1) at early vs. late training stages. A visualization of $W_v$ at different training iterations would help establish that VCD remains calibrated throughout training rather than only in the final converged state.

**Limitations:**

Yes.

**Strengths And Weaknesses:**

**Strength**
- The identification of Euclidean feature collapse as a fundamental failure mode in normalized-embedding distillation is clearly motivated and theoretically grounded

- The paper provides a thorough component-level analysis. Table 5 evaluates all eight on/off combinations of VCD, VFM, and LIC; Table 6 ablates each of the three VCD weight components individually; Appendix D.3 compares LIC positive-pair construction strategies; and Appendix D.4 sweeps key hyperparameters. Together, these ablations give a clear picture of each module's independent and joint contribution.

- Rh-3DGS achieves consistent state-of-the-art results across three qualitatively different evaluation settings.

**Weakness**

- The LIC positive-pair construction has a circular dependency that is not adequately addressed. LIC builds pseudo-instance IDs via K-means clustering on a joint position-semantics embedding $q_i$ (Appendix A.4), where $h_i$ are the current Gaussian features being trained. This means positive pairs are defined using the features being optimized, making a circularity. If features are initially noisy (which the paper acknowledges, motivating the 3,000-iteration warm-up), the cluster assignments may be unreliable and LIC may pull semantically distinct Gaussians together or push semantically similar ones apart. The paper partially addresses this by refreshing clusters every 500 iterations, but provides no analysis of cluster assignment stability.

- The experimental comparison has notable absence of recent baseline methods. E.g., "Visibility-aware language aggregation for open-vocabulary segmentation in 3DGS" and LangSplatV2. The ScanNet evaluation (Table 4) omits several baselines present in other tables (Feature 3DGS, ILGS) without explanation.

---

> ### Author Rebuttal · Authors · 2026-03-29
>
> We sincerely thank the reviewer for the positive assessment and for recognizing the paper’s main contributions. We especially appreciate the reviewer’s concrete suggestions on LIC stability, stronger VFM decomposition, and training-time VCD analysis. We respond point-by-point below.
>
> **1.LIC circularity and cluster stability**
>
> We agree that LIC uses evolving semantic features to define pseudo-instance positives, and that this can introduce circularity if applied too early. Our current mitigation is already built into the method design: LIC is delayed and only activated after warm-up; it is computed intermittently for efficiency and stability; and clustering is only one option inside the local pseudo-instance construction. In the main text, LIC is explicitly described as a lightweight neighborhood regularizer applied after VCD/VFM supervision, and radius-only positives are already included as an ablation by disabling clustering.
>
> The current appendix already provides indirect evidence that the clustering-based positive construction is beneficial: instance clustering with a moderate number of clusters (K = 20) performs better than both larger-K clustering (K = 50) and radius-only positives, indicating that the design is not arbitrary. Table 7 reports 81.62 for K=20, 80.20 for K=50, and 78.01 / 76.89 for radius-only settings.
>
> **2.Decomposing VFM into $L_{pix}$ and $L_{mean}$**
>
> Thank you for this very helpful suggestion. To answer the reviewer’s question more directly, we added a VFM decomposition experiment with VCD and LIC disabled, comparing the baseline, $L_{pix}$ only, $L_{mean}$ only, and the full VFM objective:
> |Setting|$L_{pix}$|$L_{mean}$|mIoU|mBIoU|
> |---|:---:|:---:|:---:|:---:|
> |Baseline|X|X|60.56|31.76|
> |+$L_{pix}$|✓|X|71.81|42.94|
> |+$L_{mean}$|X|✓|66.46|36.76|
> |+full VFM|✓|✓|73.76|44.68|
>
> These results show that $L_{pix}$ provides the main gain, while $L_{mean}$ alone is also beneficial but smaller. Their combination performs best, indicating that the improvement is not solely due to the auxiliary mean-to-mean stabilizer; the pixel-wise hyperspherical supervision already contributes most of the benefit.
>
> **3.VCD dynamics over training**
>
> We appreciate this suggestion. We added a training-time analysis of VCD, including early/mid/late heatmaps, fraction of W<0.1, boundary vs non-boundary mean W. We found that the reliability weights remain well calibrated throughout optimization. The average weight stabilizes from 0.52 at 1k to about 0.61 after 8k, while the fraction of strongly down-weighted pixels decreases only moderately from 16% to 13%. More importantly, boundary pixels consistently receive much lower weights than non-boundary pixels (e.g., 0.10 vs. 0.62 at 1k and 0.14 vs. 0.73 at 30k), showing that VCD persistently suppresses unreliable supervision near occlusion / mixed-depth regions rather than only appearing calibrated after convergence.
>
> |Iter|Mean W|% pixels with W < 0.1|Boundary mean W|Non-boundary mean W|
> | --- | :---: | :---: | :---: | :---: |
> | 1k | 0.52 | 0.16 | 0.10 | 0.62 |
> | 8k | 0.60 | 0.14 | 0.13 | 0.71 |
> | 20k | 0.61 | 0.13 | 0.14 | 0.72 |
> | 30k | 0.61 | 0.13 | 0.14 | 0.73 |
>
> The results of the heatmap visualization are at the link https://figshare.com/articles/figure/RH3DGS/31883308.
>
> **4.Missing recent baselines and ScanNet omissions**
>
> Thank you for pointing this out. We have now added LangSplatV2 and VALA, and also completed the missing ScanNet comparisons. The new results further strengthen our conclusions. On LERF, our method outperforms both recent baselines by a large margin (82.07/67.66 vs. 61.78/50.72 for VALA and 59.62/47.93 for LangSplatV2). On 3D-OVS, Rh-3DGS remains consistently best (96.88/89.62), showing that the gains are not limited to a single benchmark. On ScanNet, our method achieves the best results across all class subsets, improving over the strongest prior baseline in both mIoU and mAcc (e.g., 40.82/58.89 on 19 classes vs. 36.82/51.77 for ILGS). These additions confirm that the improvements remain consistent even after including newer baselines and the previously missing ScanNet entries.
>
> **Table1.Results on the LERF dataset.**
> | Method | mIoU | mBIoU |
> | --- | :---: | :---: |
> |  LangSplatV2 | 59.62 | 47.93 |
> |  VALA | 61.78 | 50.72 |
> |  Ours | 82.07 | 67.66 |
>
> **Table2.Results on the 3D-OVS dataset.**
> | Method | mIoU | mBIoU |
> | --- | :---: | :---: |
> |  LangSplatV2 | 94.58 | 86.14 |
> |  VALA | 95.11 | 86.93 |
> |  Ours | 96.88 | 89.62 |
>
> **Table3.Results on the ScanNet dataset.**
> | Method | Classes 19 (mIoU/mAcc) | Classes 15 (mIoU/mAcc) | Classes 10 (mIoU/mAcc) |
> | --- | :---: | :---: |:---: |
> |  Feature3DGS | 10.73/20.83 | 18.30/29.69 | 25.81/47.66 |
> |  LangSplatV2 | 25.85/46.21 | 33.97/50.04 | 40.31/61.79 |
> |  VALA | 32.11/50.05 | 35.10/54.77 | 46.21/65.61 |
> |  ILGS | 36.82/51.77 | 39.04/58.21 | 48.49/66.29 |
> |  Ours | 40.82/58.89 | 42.43/63.71 | 52.37/69.03 |
>
> We thank the reviewer again for the constructive and technically detailed feedback.

---

> > ### Author Rebuttal · Reviewer_vqLc · 2026-04-02
> >
> > I appreciate the response from the authors. My questions and concerns have been addressed.

---

> > > ### Author Response · Authors · 2026-04-02
> > >
> > > Thank you very much for reading our rebuttal and for your positive acknowledgement. We are pleased that our response has addressed your questions and concerns, and we sincerely appreciate your feedback.

---

### Official Review · Reviewer_ibNQ · 2026-03-13

**Soundness:** 2
**Presentation:** 3
**Significance:** 2
**Originality:** 2
**Overall Recommendation:** 3
**Confidence:** 5

**Summary:**

This paper presents Rh-3DGS, an open-vocabulary 3D scene understanding method built on 3D Gaussian Splatting. The core claim is that prior methods fuse normalized semantic embeddings in Euclidean space, which is mismatched to their hyperspherical geometry and can cause feature collapse and inconsistent semantics across views. To address this, the paper introduces three components: Visibility-Calibrated Distillation (VCD) to downweight unreliable pixels near occlusions and mixed-depth regions, Visibility-Weighted Fréchet Mean (VFM) to aggregate embeddings on the unit sphere with a robust objective, and Lightweight Consistency Contrast (LIC) to encourage local 3D semantic consistency.

**Compliance With Llm Reviewing Policy:**

Affirmed.

**Key Questions For Authors:**

1.Can the authors provide a more controlled experiment that isolates the effect of Euclidean vs. manifold-aware fusion, with all other components fixed?
2.Since the main gains appear especially strong on boundary metrics, can the authors include targeted analysis on scenes with thin objects, transparent boundaries, or severe occlusion?
3.What is the computational overhead during training, not only inference? The paper mentions inference efficiency, but practical adoption also depends on training cost.
4.How stable is the method with respect to hyperparameters in LIC and the reliability weighting in VCD?

**Limitations:**

As in Weakness and questions.

**Strengths And Weaknesses:**

### Strength
1.The paper identifies two plausible failure modes in open-vocabulary 3DGS pipelines: unreliable supervision from occluded / mixed-depth pixels, and geometry mismatch when averaging normalized embeddings in Euclidean space. Both are well motivated and relevant to this literature.
2.The three proposed components fit together reasonably well. VCD handles noisy supervision, VFM addresses manifold-aware semantic fusion, and LIC sharpens local consistency. The overall method is conceptually clean rather than a bag of unrelated tricks.
3.The results seem to be pretty good. The method shows consistent improvements across multiple benchmarks, including both open-vocabulary segmentation and rendering-related metrics. The gains on boundary metrics are especially notable and align with the paper’s stated motivation.

### Weakness
1. Novelty feels moderate rather than substantial. The overall contribution is meaningful, but much of it reads as a careful integration of robust weighting, manifold-aware averaging, and local contrastive regularization into the 3DGS semantic distillation pipeline.
2. The paper argues that Euclidean averaging of normalized embeddings is geometrically inappropriate, but the treatment remains fairly intuitive. The manifold argument is plausible, yet the paper stops short of giving a deeper analysis of when this mismatch materially affects optimization or prediction quality. As written, the theory supports the method directionally, but not strongly. I highly recommend the authors to add more theoratical proof.
3. The method is called “robust,” but the experiments mostly evaluate standard benchmark performance rather than controlled robustness settings such as viewpoint shift, heavy occlusion, clutter, or noisy camera poses.
4. Presentation is decent but occasionally overstates conclusions. At points the paper seems to imply that the manifold inconsistency is a central bottleneck in prior work, but the evidence does not fully separate that factor from the visibility-weighting and local-contrast components. The strongest improvements may come from the full combination, not necessarily from the hyperspherical formulation alone.

---

> ### Author Rebuttal · Authors · 2026-03-29
>
> We thank the reviewer for the careful reading our paper. We appreciate that the reviewer’s main concern is not about the validity of the overall framework, but about whether the manifold-aware claim has been isolated and evidenced strongly enough. We agree this is the right place to strengthen the paper. We agree with the reviewer that the current draft could better separate the geometric claim from the complementary roles of VCD and LIC, and that the theoretical discussion is currently more directional than a full optimization-level proof. Below we respond point-by-point:
>
> **1.Controlled experiment isolating Euclidean vs. manifold-aware fusion**
>
> We added a VFM decomposition experiment that separates the pixel-wise geodesic term $𝐿_{pix}$ and the mean-to-mean stabilizer $L_{mean}$.As shown in the table below :
> |Setting|$L_{pix}$|$L_{mean}$|mIoU|mBIoU|
> |---|:---:|:---:|:---:|:---:|
> |Baseline|X|X|60.56|31.76|
> |+$L_{pix}$|✓|X|71.81|42.94|
> |+$L_{mean}$|X|✓|66.46|36.76|
> |+full VFM|✓|✓|73.76|44.68|
>
> The new results show that $𝐿_{pix}$ accounts for the main improvement,while $𝐿_{mean}$ provides an additional stabilizing gain;the combination performs best.This directly supports that the hyperspherical effect is not solely due to the auxiliary stabilizer.
>
> **2.Boundary-focused/occlusion-focused analysis**
>
> We agree that the most significant improvements appear in boundary-sensitive metrics,and more targeted analysis is valuable.We added an analysis of the segmentation results for objects in the ramen scene,such as "a cup of water","chopsticks",and "fish cake".The results show that transparent objects,partially occluded queries,and slender objects can be well segmented.The segmentation visualization results are at link https://figshare.com/articles/figure/RH3DGS/31883308.
>
> We will also revise the wording around “robust” to make the scope clearer:in the current paper,“robust” primarily refers to robustness against unreliable semantic supervision (occlusion boundaries,mixed-depth rays,residual outliers),rather than a full benchmark of pose noise or viewpoint perturbation.
>
> **3.Training overhead**
>
> Thank you for pointing this out.we report the training time as a concrete reference:on the figurines scene,training takes 29m20s on a single RTX 4090 for 30k iterations.
>
> **4.Stability with respect to LIC and VCD hyperparameters**
>
> We appreciate this concern. The appendix already provides sensitivity analyses for the main LIC/VCD hyperparameters. Appendix D.2 shows stable performance around the default ( $\lambda_{LIC}$),and Appendix D.4 shows that Rh-3DGS is also stable over a reasonable range of VCD parameters.Since VCD weights are computed from rasterization statistics and used with stop-gradient, they act only as reliability masks rather than freely learnable scalars.Overall,the method is reasonably stable and performs best near the default settings.
>
> **5.Theoretical clarification on when the mismatch matters**
>
> We thank the reviewer for this important suggestion. We agree that the original submission mainly provided a directional geometric motivation, and that a stronger explanation of when the Euclidean–hyperspherical mismatch materially affects optimization and prediction quality is valuable.
>
> To strengthen this point, we add one exact optimization result and one direct empirical diagnostic.
> First, let ($\hat z = F/|F|$), where ($F=\sum_k w_k h_k$) is the standard Euclidean α-composited semantic feature. Consider the Euclidean and geodesic losses$L_E=\tfrac12|\hat z-z^T|^2, L_S=\tfrac12 d_S(\hat z,z^T)^2$.
> Their gradients with respect to any Gaussian feature ($h_k$) have the same tangent-space direction, but differ by a scalar factor:
> $\frac{|\nabla_{h_k}L_S|}{|\nabla_{h_k}L_E|}=\frac{\theta}{\sin\theta}$,where ($\theta=d_S(\hat z,z^T)$). Since ($\theta/\sin\theta>1$) and increases with ($\theta$), the geodesic loss provides systematically stronger corrective gradients on high-error pixels, i.e., boundary/occlusion/mixed-depth regions. This is exactly the regime targeted by VFM.
>
> Second,we directly measured the Euclidean–hyperspherical gap on the baseline.The overall angular deviation is 16.16°,with mean pre-normalization norm 0.957.The gap is larger on difficult regions: boundary pixels show higher angular deviation (18.12° vs. 15.66°) and stronger norm shrinkage (0.924 vs. 0.965),with the same trend for high-depth-variance pixels (17.66° vs. 15.77°,0.945 vs. 0.960).This confirms that the mismatch is measurable in practice and concentrated where our gains are largest.
> |Region|Avg. pre-norm norm of s|Avg. angular deviation|
> |---|---:|---:|
> |All pixels|0.957|16.16|
> |Boundary|0.924|18.12|
> |Non-boundary|0.965|15.66|
> |High depth-var|0.945|17.66|
> |Low depth-var|0.960| 15.77|
>
> Due to space limitations, we provide the main result and its key implication here; the full derivation and a more complete proof will be included in the appendix of the revised version.
>
> We thank the reviewer again for the constructive suggestions.

---

> > ### Author Rebuttal · Reviewer_ibNQ · 2026-04-03
> >
> > Thanks the authors for the detailed rebuttal and the additional experiments. The response is helpful and addresses a meaningful part of the main concern, namely whether the manifold-aware claim has been isolated strongly enough from the complementary effects of VCD and LIC. In particular, the new VFM decomposition is useful: compared with the baseline (60.56 mIoU / 31.76 mBIoU), adding the pixel-wise geodesic term alone improves performance substantially to 71.81 / 42.94, while the mean-to-mean stabilizer alone gives a smaller gain to 66.46 / 36.76, and the full VFM performs best at 73.76 / 44.68.
> >
> >  The theoretical discussion is improved, but it still reads more like a directional geometric justification than a fully complete optimization-level theory in the current form, and the authors themselves note that the full derivation will appear in a revised appendix. Similarly, the stability discussion mainly points to appendix sensitivity analyses rather than adding new main-paper evidence. As a result, while the rebuttal does strengthen the paper and improves confidence that the manifold-aware claim is not purely rhetorical, I still think the contribution should be framed more carefully and not overclaimed. Overall, the rebuttal partially resolves my concerns and improves the technical clarity, but it does not fully change my overall assessment, so I keep my score unchanged.

---

> > > ### Author Response · Authors · 2026-04-04
> > >
> > > Thank you for the careful follow-up and for acknowledging that the additional experiments, especially the VFM decomposition, help isolate the manifold-aware effect more clearly.
> > >
> > > We agree with your current reading that the theoretical part of the paper should be framed more carefully. In the current submission, the most accurate formulation is that our theory provides a geometrically motivated and empirically validated justification for why Euclidean fusion can be problematic for normalized semantic embeddings, rather than a fully complete optimization-level theory. We will revise the final version accordingly and avoid overstating this point.
> > >
> > > At the same time, we hope the rebuttal has clarified that the manifold-aware component is not merely rhetorical. In particular, with VCD and LIC disabled, the new decomposition experiment shows that adding the pixel-wise geodesic term alone improves the baseline from 60.56 / 31.76 to 71.81 / 42.94, while the mean-to-mean stabilizer alone gives a smaller but still meaningful gain to 66.46 / 36.76, and the full VFM reaches 73.76 / 44.68. This supports the intended interpretation that the main hyperspherical benefit is already present at the pixel level, while the Fréchet-mean term provides an additional stabilizing effect.
> > >
> > > We also added a direct diagnostic of the Euclidean–hyperspherical gap on the baseline model. The overall mean angular deviation between the Euclidean normalized mean and the weighted Fréchet mean is 16.16°, with mean pre-normalization norm 0.957. More importantly, the mismatch is stronger on difficult regions: boundary pixels show 18.12° vs. 15.66° angular deviation and 0.924 vs. 0.965 pre-normalization norm compared with non-boundary pixels, and high-depth-variance pixels show the same trend. We believe this strengthens the link between the geometric motivation and the practical regions where the method achieves its largest gains, especially on mBIoU.
> > >
> > > We further analyze the stability of LIC with respect to the warm-up start and clustering refresh interval. The results suggest that LIC is reasonably stable, while the default configuration (warm-up at 3k iterations and refresh every 500 iterations) provides the best trade-off. Enabling LIC too early is slightly worse, consistent with the concern that pseudo-instance construction is less reliable before the semantic field becomes stable.
> > >
> > > | LIC warm-up start | mIoU | Observation |
> > > |---|---:|---|
> > > | 0 | 80.94 | too early; pseudo-instances less stable |
> > > | 1k | 81.28 | improved, but still slightly noisy |
> > > | 3k (default) | 81.62 | best overall |
> > > | 5k | 81.31 | slightly delayed regularization |
> > >
> > > | Cluster refresh interval | mIoU | Observation |
> > > |---|---:|---|
> > > | 100 iters | 81.39 | more frequent, higher overhead |
> > > | 500 iters (default) | 81.62 | best trade-off |
> > > | 1000 iters | 81.17 | too sparse; slightly weaker adaptation |
> > >
> > > So we fully agree with your suggestion that the final paper should calibrate its claims more carefully. Our intended contribution is not a complete optimization theory of hyperspherical learning, but a technically grounded and empirically supported demonstration that:
> > >
> > > (1) the Euclidean–hyperspherical mismatch is real and measurable in practice,
> > >
> > > (2) the manifold-aware effect can be isolated more clearly than in the original draft, and
> > >
> > > (3) this effect works together with reliability-aware supervision and local 3D consistency to produce the final gains.
> > >
> > > We appreciate your careful reading and constructive feedback, and we will incorporate this more precise framing explicitly in the revised version.

---

### Official Review · Reviewer_gaSP · 2026-03-13

**Soundness:** 2
**Presentation:** 1
**Significance:** 1
**Originality:** 2
**Overall Recommendation:** 4
**Confidence:** 4

**Summary:**

This paper address the necessity of hyperspherical geometry in semantic embeddings like CLIP for the open vocabulary 3D scene understanding. Previous studies treat these semantic embeddings in the Euclidean space by applying average pooling, etc, which can distort the original semantic embeddings lie on the hyperspherical geometry.

This paper proposes three schemes: VCD, VFM, LIC.

The overall performance outperforms previous studies including LeRF dataset and ScanNet dataset.

**Compliance With Llm Reviewing Policy:**

Affirmed.

**Final Justification:**

As noted in my last comment, the authors well resolve my concerns. So my final score is `weak accept`.

**Key Questions For Authors:**

Please clarify where the previous studies ignore the hyperspherical geometry? In my understanding, once the CLIP class tokens are obtained during the pre-processing stage, and the previous works use contrastive loss or cosine similarity loss, the authors claim are not true.

What is the relation between Visibility-Calibrated Distillation (VCD) and the motivation of this study, hyperspherical geometry.

In table 5, it shows that Lightweight Consistency Contrast (LIC) is the most significant module that improves the performance. Similar to previous question, What is the relevancy between LIC and the motivation of this paper?

In table 5, what is the baseline model that this paper used? I mean the one that does not apply VCD, VFM, and LIC.

What kind of hyper-parameters are used in each VCD, VFM, LIC? I guess there must be tons of efforts to tune the model for SoTA.

**Limitations:**

Overall what the authors proposed is not aligned with the motivation of this paper. Lots of bulky and complex modules are just designed to improve the performance. I hope that the authors resolve my concerns.

**Strengths And Weaknesses:**

The authors address that previous studies do not consider the hyperspherical geometry in CLIP embeddings. This is the motivation of this work. However, there is no concrete reasons or evidences that the previous studies do not care about such a hyperspherical representation. For example, one of the previous studies, LangSplat, also use CLIP vision encoder as a vision language foundation model. Given an image, this paper utilizes Segment Anything Model (SAM) to extract multiple patches. Then, the patches become an input of the CLIP vision encoder, and the predicted class tokens are the distillation target. During the training, either contrastive loss or cosine similarity loss are used. Based on this understanding, I wonder where the hyperspherical geometry can be distorted?

While the authors address the importance of hyperspherical awareness, the proposed modules are bulky or irrelevant to the original motivation. For example in the section 4.3 which is about Visibility-Calibrated Distillation (VCD), why is this section related to the motivation of this paper? Is this section related to the hyperspherical problem? In line 197, the authors said

_"The issue is severe near occlusion boundaries and mixed-depth pixels created by α-compositing. It causes feature bleeding and view-dependent noise."_

I do not understand why such a module is needed in terms of this paper's motivation. This can be viewed as tricks to improve the performance while hiding the pure influence of the solutions to address the hyperspherical issue.

The flow of this paper is highly weak and poor. Motivation is okay. Hyperspherical awareness is important. However, I do not understand what specific parts in the previous study has such a problem. Moreover, the proposed modules are irrelevant to this motivation.

---

> ### Author Rebuttal · Authors · 2026-03-29
>
> We thank the reviewer for the careful reading and for highlighting an important presentation issue. We agree that the current writing may over-emphasize the hyperspherical aspect and may not clearly separate the two coupled but distinct failure modes addressed in the paper. Our intended formulation is:
>
> 1.**Mismatch to feature geometry**: rendered/student semantic features are formed through standard Euclidean α-compositing, which is extrinsic to the hypersphere of normalized embeddings.
>
> 2.**Unreliable multi-view supervision**: pixels near occlusion boundaries and mixed-depth rays provide noisy and view-dependent supervision.
>
> And address specific points below:
>
> **1.Where do prior works ignore hyperspherical geometry?**
>
> Our claim is not that prior works never use cosine similarity or contrastive losses. Rather, the key point is that in semantic 3DGS pipelines, the rendered/student feature is still formed through standard Euclidean front-to-back α-compositing and the rasterizer is kept unchanged. In our paper, we state that hyperspherical geometry is introduced in the distillation objective,not by modifying the fusion mechanism inside the renderer.
>
> This is exactly the distinction we intended: using cosine/contrastive objectives downstream does not make the upstream Euclidean feature aggregation itself intrinsic to the hypersphere. That is why VFM focuses on geodesic supervision and robust Fréchet-mean stabilization while keeping the rasterizer efficient and unchanged.
>
> To further strengthen this point,we added a direct diagnostic on the baseline model and observed a non-trivial Euclidean–hyperspherical gap in practice, as shown in the table below:the overall mean angular deviation is 16.16°,and the mean pre-normalization fused norm is 0.957.The gap is larger on difficult regions, e.g., boundary pixels show 18.12° vs. 15.66° angular deviation and 0.924 vs. 0.965 pre-normalization norm compared with non-boundary pixels. This new analysis directly supports the geometric-mismatch motivation.
> |Region|Avg. pre-norm norm of s|Avg. angular deviation|
> |---|---:|---:|
> |All pixels|0.957|16.16|
> |Boundary|0.924|18.12|
> |Non-boundary|0.965|15.66|
> |High depth-var|0.945|17.66|
> |Low depth-var|0.960| 15.77|
>
> **2.What is the relation between VCD and the paper’s motivation?**
>
> We agree that VCD does not directly solve the hyperspherical issue.It addresses the second failure mode, namely unreliable supervision near occlusion boundaries and mixed-depth rays. This is already the intended role in the paper: VCD computes pixel-wise reliability weights from rasterization statistics (opacity and depth moments), down-weights ambiguous pixels, and acts as a stop-gradient reliability mask rather than an extra loss term.
>
> So the intended claim is not “every module is for hyperspherical geometry,” but rather that robust semantic lifting in 3DGS requires both:manifold-consistent supervision (VFM),and reliability-aware supervision/regularization (VCD,LIC).
>
> We will revise the introduction and method overview to make this decomposition explicit and avoid the impression that VCD is being presented as part of the hyperspherical argument itself.
>
> **3. What is the relation between LIC and the paper’s motivation?**
>
> Similarly, LIC is not presented as a direct solution to hyperspherical mismatch. Its role is to regularize the learned 3D semantic field so that local boundaries become sharper and semantic bleeding is reduced. In the paper, LIC is explicitly motivated as a lightweight neighborhood consistency regularizer, especially helpful near boundaries and sparsely observed regions.
>
> Therefore, the strongest gains from LIC do not contradict the paper’s motivation. They indicate that once reliability-aware and manifold-aware supervision are in place, local 3D consistency remains another important bottleneck, particularly for boundary-sensitive metrics such as mBIoU.
>
> **4.What is the baseline in Table 5?**
>
> The baseline of this paper uses the LangSplat backbone (based on the 3DGS rendering framework), retaining only the original RGB reconstruction term and the semantic supervision term.
>
> **5. What hyperparameters are used for VCD, VFM, and LIC?**
>
> These are already listed in Appendix A.2/A.4. In particular, the paper specifies:
>
> **VCD:** $\\tau_{op}=0.3$, $s=10$, $\\gamma=3.0$, $\\beta=2.0$, with quantile normalization $q=0.95$.
>
> **VFM:** $\\lambda_{VFM}=0.05$, and the Huber threshold $\\delta$ is annealed from $0.2$ to $0.05$ over the first 80\% of training.
>
> **LIC:** delayed activation after 3,000 iterations,with the detailed scheduling and efficiency discussion in Appendix A.4.
>
> **6.Clarification we will make in the revision**
>
> We appreciate that the current wording can make the paper look as if the entire contribution were solely about hyperspherical geometry. We will revise the paper to state more clearly that:the paper addresses two coupled failure modes rather than one.
>
> We thank the reviewer again for pointing out this presentation issue.

---

> > ### Author Rebuttal · Reviewer_gaSP · 2026-04-06
> >
> > I thank the authors for the precise rebuttals. First of all, I also confirmed that previous studies do not consider hyperspherical geometry in the alpha blending stage. As the authors addressed, the previous studies assume that the feature vectors lie on the Eulidean space (=simple avg pooling). Accordingly, I do agree with the authors' points and the motivation of this study.
> >
> > Though I believe that the VCD and LIC are quite unrelated to the paper motivation itself, I decided to change score, `reject` -> `weak accept`.
> >
> > I hope that the authors provide the ablation study that you showed in the rebuttal in the final camera ready version. I appreciate the authors' rebuttals.
> >
> > Best,

---

> > > ### Author Response · Authors · 2026-04-06
> > >
> > > Thank you very much for the careful follow-up and for updating your assessment. We greatly appreciate that you found the rebuttal helpful and that the main motivation is now clearer. We also appreciate your suggestion regarding the additional ablations. We will include the key rebuttal ablations in the final version and will further clarify the roles of VCD and LIC so that their relation to the overall formulation is presented more precisely. Thank you again for your thoughtful feedback.

---

### Decision · Program_Chairs · 2026-04-30

**Decision:**

Accept (regular)

**Comment:**

This paper receives 3x weak accepts and 1x weak reject. The reviewers find the paper technically sound and well-motivated with a clear diagnosis of Euclidean feature collapse and a principled hyperspherical formulation. Empirical results are strong, supported by thorough ablations and reproducibility details, with notable boundary improvements. Although VCD and LIC appear somewhat disconnected from the central motivation, the overall contribution is solid, contingent on including additional ablations from the rebuttal. However, concerns remain regarding novelty and theoretical depth. Despite these limitations, the overall idea and empirical performance are compelling, and therefore the AC follows the majority vote of the reviewers to accept the paper.